# Integrating multi-omics data reveals function and therapeutic potential of deubiquitinating enzymes

Laura M Doherty[1,2,3,4†], Caitlin E Mills[4], Sarah A Boswell[4‡], Xiaoxi Liu[2,3], Charles Tapley Hoyt[4], Benjamin Gyori[4], Sara J Buhrlage[2,3]*, Peter K Sorger[4]*

[1]Harvard Medical School (HMS) Library of Integrated Network-based Cellular Signatures (LINCS) Center, Cambridge, United States; [2]Department of Cancer Biology and the Linde Program in Cancer Chemical Biology, Dana-Farber Cancer Institute, Boston, United States; [3]Department of Biological Chemistry and Molecular Pharmacology, Harvard Medical School, Boston, United States; [4]Laboratory of Systems Pharmacology, Department of Systems Biology, Harvard Program in Therapeutic Science, Harvard Medical School, Boston, United States

**\*For correspondence:**
saraj_buhrlage@dfci.harvard.edu (SJB);
peter_sorger@hms.harvard.edu (PKS)

**Present address:** †Laura Doherty, Broad Institute, Cambridge, United States; ‡Sarah Boswell, Ginkgo Bioworks, Boston, United States

**Abstract** Deubiquitinating enzymes (DUBs), ~100 of which are found in human cells, are proteases that remove ubiquitin conjugates from proteins, thereby regulating protein turnover. They are involved in a wide range of cellular activities and are emerging therapeutic targets for cancer and other diseases. Drugs targeting USP1 and USP30 are in clinical development for cancer and kidney disease respectively. However, the majority of substrates and pathways regulated by DUBs remain unknown, impeding efforts to prioritize specific enzymes for research and drug development. To assemble a knowledgebase of DUB activities, co-dependent genes, and substrates, we combined targeted experiments using CRISPR libraries and inhibitors with systematic mining of functional genomic databases. Analysis of the Dependency Map, Connectivity Map, Cancer Cell Line Encyclopedia, and multiple protein-protein interaction databases yielded specific hypotheses about DUB function, a subset of which were confirmed in follow-on experiments. The data in this paper are browsable online in a newly developed DUB Portal and promise to improve understanding of DUBs as a family as well as the activities of incompletely characterized DUBs (e.g. USPL1 and USP32) and those already targeted with investigational cancer therapeutics (e.g. USP14, UCHL5, and USP7).

## Editor's evaluation

This study reports the creation of a database on deubiquitinating enzymes (DUBs), which integrates existing large-scale datasets with new knock-out and inhibition experiments. The combined data confirm known DUB functions and, importantly, correct several current assumptions and highlight potential new functions of DUBs. The data are made available through an online portal, providing a useful resource for investigators interested in DUB functions or considering DUBs as drug targets.

## Introduction

Deubiquitinating enzymes (DUBs) are a family of ~100 proteases (in humans) that cleave ubiquitin from protein substrates (*Komander et al., 2009*). They are essential components of the ubiquitin-proteasome system (UPS), which regulates protein turn-over in cells by tagging polypeptide substrates with poly-ubiquitin chains. These poly-ubiquitin chains involve linkages between the C terminus of one ubiquitin molecule (of 76 amino acids) and one of seven lysine residues or N-terminal methionine on

the next ubiquitin molecule. Lysine 48-linked chains are among the ones recognized by the proteasome, resulting in degradation of the substrate. The primary function of DUBs in this process is to remove ubiquitin molecules from substrates, thereby protecting them from proteasomal degradation (*Nandi et al., 2006*). However, ubiquitination can also regulate protein localization, enzyme activity, and recruitment of binding partners; in many cases, these types of regulation involve monoubiquitin adducts or ubiquitin chains linked to the substrate and each other via a lysine residue other than K48 (e.g. Lysine 63) but these too can be removed by DUBs. Thus, DUBs can regulate multiple cellular processes other than protein degradation (*Kerscher et al., 2006*).

A growing body of literature shows that DUBs are dysregulated in many disease settings including cancer, chronic inflammation, and neurodegenerative diseases (*Popovic et al., 2014*; *Park et al., 2014*; *Atkin and Paulson, 2014*; *Shi and Grossman, 2010*) and that DUBs may be useful targets for the development of therapeutic drugs (*Kerscher et al., 2006*; *Komander and Rape, 2012*). Inhibiting DUBs with small molecules has emerged as a particularly promising means of indirectly targeting proteins that are conventionally considered to be 'undruggable', typically due to the absence of a binding pocket into which a small molecule might bind; such proteins include transcription factors and scaffolding proteins (*Dang et al., 2017*). For example, USP7 is a DUB that stabilizes MDM2, the E3 ligase for the tumor suppressor TP53, and inhibiting USP7 has emerged as a strategy for indirectly increasing the levels of TP53, which is among the most highly mutated genes in cancer but has thus far eluded direct targeting by small molecules (*Schauer et al., 2020*). Similarly, USP28 is a DUB that stabilizes the c-Myc transcription factor, a potent oncogene in a wide variety of human cancers, and inhibiting USP28 is expected to reduce the levels of c-Myc, and downregulate its activities (*Weisberg et al., 2017*; *Wrigley et al., 2017*; *Zhou et al., 2018*). However, effectively exploiting DUB biology to upregulate tumor suppressor proteins such as TP53 and downregulate oncogenes such as c-Myc requires a more complete understanding of DUB specificity, regulation, and knock-down phenotypes.

Only a few DUB inhibitors have entered clinical testing, and none are as-yet approved (*Antao et al., 2020*). The current clinical pipeline includes small molecules targeting USP1 (with a phase 1 clinical trials underway sponsored by KSQ Therapeutics, Inc in advanced solid tumors) and USP30 (with clinical trial plans announced by Mission Therapeutics for kidney disease) (*KSQ Therapeutics, Inc, 2022*, Identifier: NCT05240898; *Mission Therapeutics, 2022*). While potent and selective inhibitors have been described for a small number of DUBs, including USP7, USP14, and CSN5 (*Schauer et al., 2020*; *Schlierf et al., 2016*; *Wang et al., 2018a*), development of chemical probes for other DUBs has largely yielded relatively non-selective compounds (*Ndubaku and Tsui, 2015*). Moreover, many potential DUB substrates reported in the literature have not yet been explored in detail or fully confirmed, and the dependency of DUB-substrate interactions on biological setting (e.g. cell type or state) is largely unexplored. Lastly, out of more than 100 DUBs in the human proteome, only a small subset has been studied from a functional perspective. This lack of information on DUBs as a family makes the task of prioritizing DUBs for development of chemical probes and, ultimately, human therapeutics, more difficult.

In this paper, we take a combined and relatively unbiased computational and experimental approach to investigating the functions of DUBs, with an emphasis on cancer cell lines in which DUB inhibitors have been most extensively studied. First, we use high-throughput RNA-seq to measure the transcriptome-wide impact of an 81-member DUB CRISPR-Cas9 knockout library and also of seven small molecule DUB inhibitors chosen for high selectivity. As a complementary source of phenotypic and molecular information, we mine several large omics datasets (*Table 1*), including the Dependency Map (DepMap) (*Meyers et al., 2017*; *Tsherniak et al., 2017*), the Connectivity Map (CMap) (*Lamb et al., 2006*) and a recently published Cancer Cell Line Encyclopedia (CCLE) proteomics dataset (*Nusinow et al., 2020*) as well as multiple protein-protein interaction databases (PPIDs) (*Cerami et al., 2011*; *Hermjakob et al., 2004*; *Malovannaya et al., 2011*; *Stark et al., 2006*). We then test a subset of the inferred DUB activities using focused knockout and functional studies.

The DepMap aims to identify genes that are essential for cell proliferation based on a genome-wide pooled CRISPR-Cas9 knockout screen conducted in more than 700 cancer cell lines spanning multiple tumor lineages. Each cell line in the DepMap carries a unique barcode to enable parallel analysis, and each gene knockout is assigned a 'dependency score' on a per cell-line basis. The dependency score quantifies the rate at which a cell carrying a particular CRISPR-Cas9 guide RNA is outcompeted ('drops out') in a specific cell line; the more negative the dependency score, the stronger

**Table 1.** Data and public resources used to infer the functions of DUB genes.

| Resource | Description | Key Insights | Terms / abbreviations |
|---|---|---|---|
| DGE RNA-seq | High-throughput RNA-seq data collected for this study following knockout or inhibition of individual DUBs using CRISPR-Cas 9. 81 DUB knockouts (found in a commercial library) and 7 small molecules were characterized in biological replicate. | Transcriptional signatures acquired from cells in which individual DUBs are inactivated. This provides insight into potential gene function. Similarity between signatures of gene knockout and small molecules provides insight into small molecule selectivity. | *3'DGE-seq*: 3' Digital Gene Expression, a type of high-throughput RNA-seq. *Differentially Expressed (DE) gene*: Gene with different level of expression across two samples as defined by a false discovery rate (FDR) adjusted *P*<0.05. |
| Connectivity Map (CMap) | A Broad Institute database of post-perturbation RNA-seq signatures generated from multiple cell lines following knockdown (RNAi or CRISPR-Cas 9), gene over-expression, or treatment of cells with small molecule drugs. Signatures in CMAP resource are comprised of 978 landmark genes measured using a Luminex bead-based assay. The expression of 11,350 genes is then inferred. Data are available for ~3000 genes and ~5000 small molecules. | Enables identification of genes that, when silenced with RNAi, or overexpressed, have similar transcriptional effects as a query L1000 or DGE-Seq signature. The effects of gene knockout/over-expression can be compared to the effects of drugs for mechanism of action studies. | *Query mRNA profile*: the transcriptomic signature that is used to query CMap and retrieve similar signatures. *Tau score*: a parameter that quantifies similarity between the query mRNA profile and CMap signatures (tau similarity is computed by counting the number of pairwise mismatches between two ranked lists). CMap recommends a threshold value for tau similarity scores of >90. |
| Dependency Map (DepMap) | A Broad Institute database of gene essentiality scored in >700 cancer cell lines based on genome-wide pooled CRISPR-Cas9 knockout screens. Dropout of specific Guide RNAs is used as a measure of essentiality. | Enables identification of genes that are essential for cell proliferation or survival in specific cell lines. Patterns of essentiality across cell line panels (the DepMap score) can be computed to identify genes potentially having related biological functions. | *DepMap score* – a measure of cell line dropout rate in a pooled genome-wide CRISPR screen (a gene with a dependency score <–0.5 is considered an essential gene in that cell line) *Co-Dependency*. Genes with similar DepMap scores are said to be *co-dependent*. In our study, we analyzed the top seven co-dependent genes, but similar data were obtained when more or fewer co-dependent genes were considered. |
| Cancer Cell Line Encyclopedia (CCLE) Proteomics | A Broad Institute database of baseline shotgun proteomics collected from CCLE cell lines. Data from ~375 cell lines and 12,000 proteins per line are available. | Protein co-expression across cell line panels provides insight into functional interactions: proteins in the same complex are often co-expressed to a significant degree across CCLE cell clines. | *Co-expressed genes*: correlation in protein abundance across cell lines with FDR <0.01 and |z-score|>2. These thresholds are set based on previous publications (*Nusinow et al., 2020*). |
| BioGRID | Protein-protein interaction database compiling interaction data from multiple sources. Protein interactions are measured using multiple physical assays including affinity capture MS, affinity capture western blotting, and assembly of reconstituted complexes from purified recombinant subunits in vitro. | Discovery of protein-protein interactions using a variety of methods that focus on physical interaction. | *PPID:* protein-protein interaction database. |
| IntAct | Protein-protein interaction database that compiles interaction data from multiple sources. | Discovery of protein-protein interactions using a variety of methods that focus on physical interaction. | *PPID:* protein-protein interaction database. |
| Pathway Commons | Protein-protein interaction database compiling interaction data from multiple sources. | Discovery of protein-protein interactions using a variety of methods that focus on physical interaction. | *PPID:* protein-protein interaction database. |
| NURSA | Physical interactions among proteins, particularly those involved in transcription. | Discovery of proteins that interact directly as determined by affinity capture mass spectrometry; approximately ~3000 IP assays are currently included. | *PPID:* protein-protein interaction database. *NURSA*: Nuclear Receptor Signaling Atlas |

the impact on proliferation and thus, the higher the rate of guide disappearance from the population (*Meyers et al., 2017*; *Tsherniak et al., 2017*). The dependency score therefore provides a measure of the essentiality of a gene in different cell lines.

The CMap is a database of ~1000-gene mRNA signatures obtained from cells treated with small molecule drugs or in which individual genes have been knocked down using RNAi or overexpressed; data are available for ~3000 different genes and ~5000 small molecules. Each mRNA signature involves the measurement of a representative subset of the transcriptome using a bead-based (Luminex) assay (*Lamb et al., 2006*). The CCLE proteomics dataset is comprised of shotgun proteomic data for 375 cell lines from multiple tumor lineages without perturbation; ~12,000 proteins are detected in total across the dataset (*Nusinow et al., 2020*).

PPI datasets were obtained from BioGRID, IntAct, and Pathway Commons PPIDs, as well as the NURSA dataset that is focused on interactions among proteins involved in transcription; in these datasets, interaction was assessed using a variety of methods including affinity capture followed by mass spectrometry, affinity capture followed by western blotting, and assembly of reconstituted

complexes from purified recombinant subunits (*Cerami et al., 2011*; *Hermjakob et al., 2004*; *Malovannaya et al., 2011*; *Rouillard et al., 2016*). When combined with CRISPR screens and focused hypothesis-testing experiments, data mining provided new insight into the functions and interactors of the majority of human DUBs. These data set the stage for further analysis of the DUB protein family and for development of chemical probes for specific DUBs and DUB subfamilies.

## Results

### General description of integrative multi-omics approach

To characterize the DUB family of enzymes, we combined laboratory experiments and data mining (*Figure 1*, *Table 1*). As a first step, we leveraged CMap to identify genes that, when silenced with RNAi, or overexpressed, had similar transcriptional effects as DUB knockouts. This approach is based on the observation that genetic perturbations acting on components of the same or related pathways frequently give rise to significantly similar transcriptomic signatures (*Lamb et al., 2006*). To generate knockouts, we used a commercially available arrayed CRISPR-Cas9 library targeting 81 DUBs and 13 additional proteins in the ubiquitin-proteasome system, including ubiquitin-like proteins; the library was constructed with four, pooled, guides per target (coverage of DUBs is incomplete in this CRISPR-Cas9 library because it was based on a historical understanding of the DUB family; *Figure 2a*, *Supplementary file 1*). Prior to screening, transfection was optimized by assaying the abundance of selected target proteins using western blots (see Materials and methods, *Figure 2—figure supplement 1a*). mRNA profiling was performed 96 hr after guide RNA transfection using a high-throughput, low-cost RNA-sequencing method (3' Digital Gene Expression; 3'DGE-seq) in the MDAMB231 breast cancer cell line (*Figure 2b*, *Semrau et al., 2017*; *Soumillon et al., 2014*). Four days after guide RNA transfection, we generated mRNA profiles by 3'DGE-seq and then queried the CMap database. This yielded tau scores quantifying similarity between the query mRNA profile and CMap signatures (tau similarity is computed by counting the number of pairwise mismatches between two ranked lists) (*Lamb et al., 2006*). We used the recommended threshold of tau similarity score >90 to determine significantly similar perturbations.

Next, we leveraged the DepMap dataset to investigate DUB essentiality. It has been observed that genes having similar DepMap scores across large panels of cell lines are likely to have related biological functions (*Meyers et al., 2017*; *Pan et al., 2018*; *Tsherniak et al., 2017*), a property known as co-dependency. More specifically, co-dependent genes are frequently found to lie in the same or parallel pathways (as defined by gene ontology (GO), for example) or to be members of the same protein complex. We identified co-dependent genes for DUBs and then ran GO enrichment analysis on the co-dependent genes to identify pathways in which the DUBs were likely to be active. We also mined data on co-dependent genes on four other datasets. First, we asked whether co-dependent genes had similar transcriptomic signatures in CMap. Second, we used protein-protein interaction databases (PPIDs) such as BioGRID, IntAct, Pathway Commons, and NURSA to ascertain whether co-dependent genes might interact physically with one another. Third, we searched CCLE proteomics data for proteins whose expression levels across ~375 cell lines strongly correlated with the level of each DUB; it has previously been observed that proteins in the same complex are often co-expressed to a significant degree across a cell line panel (*Nusinow et al., 2020*). Fourth, we repeated GO-enrichment analysis for protein-interactors identified from PPIDs or for significantly co-expressed proteins from the CCLE proteomics data. All four approaches involve indirect assessment of function or interaction but they are based on different types of data and different ideas about what constitutes a gene-gene 'interaction'. We reasoned that inferences that were significantly correlated or at least consistent across data sources were more likely to be biologically meaningful. As one means of testing this hypothesis we asked whether results for well-studied DUBs were consistent with prior literature knowledge.

Experimental support for functional associations obtained from data mining was sought by knocking out the DUB and several co-dependent genes and then assaying phenotypic similarity by mRNA profiling and other means. We also compared CRISPR-Cas9 knockout phenotypes with phenotypes induced by treatment of MDAMB231 or MCF7 cells with one of seven recently developed small molecule DUB inhibitors. We selected these inhibitors based on their reported selectivity for specific

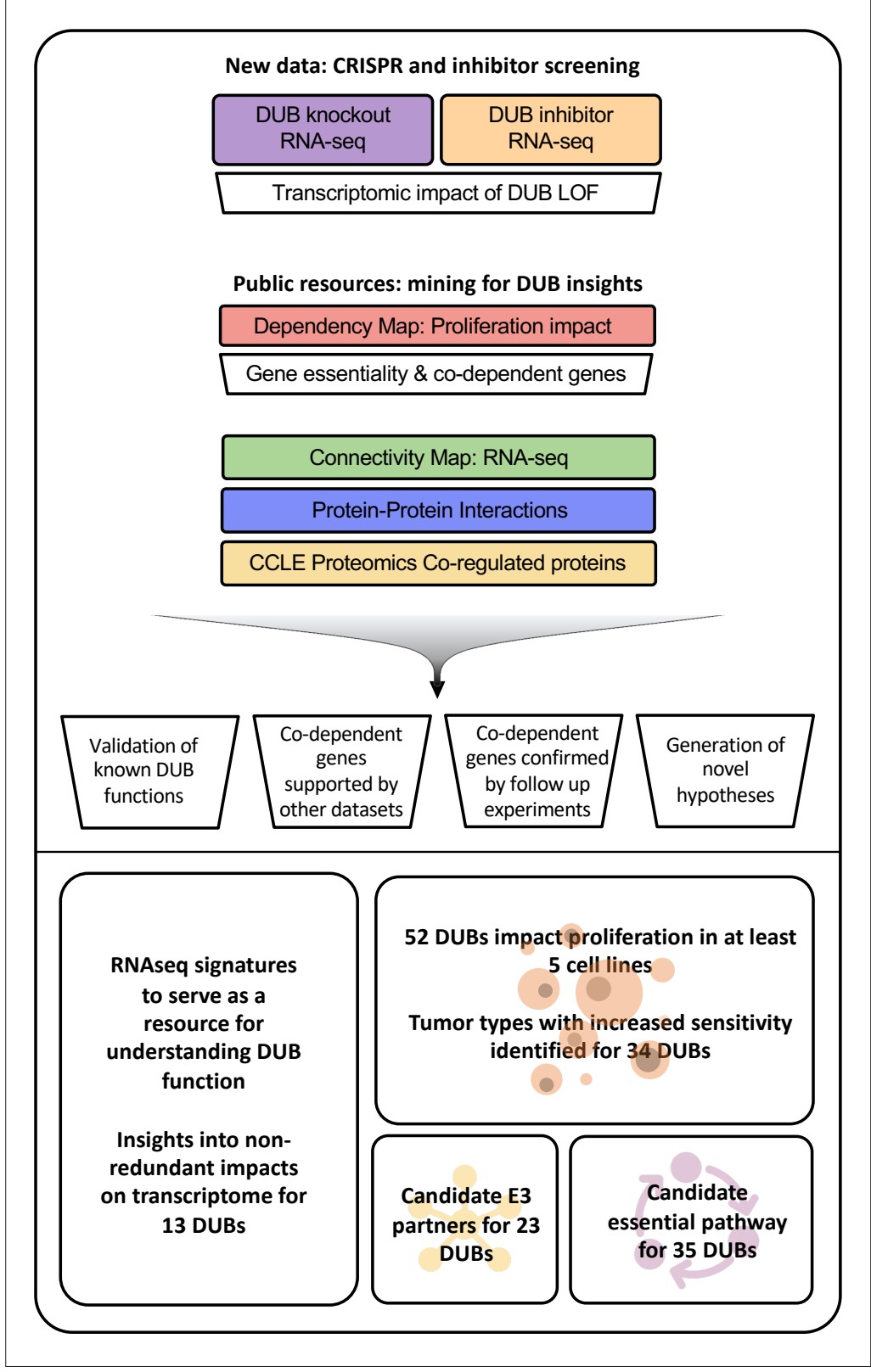

**Figure 1.** Approach to multi-omics analysis DUBs. The functional impact of DUB loss was investigated via transcriptomic profiling of cells following CRISPR-Cas9 knockout (purple box); this was compared to signatures for genetic perturbations in the Broad Connectivity Map database of RNA-seq signatures (green). The impact of DUB knockout on cancer cell line proliferation was analyzed using the Broad Dependency Map (red) and

*Figure 1 continued on next page*

*Figure 1 continued*

compared to other gene knockouts in the dataset; this identified co-dependent genes (sets of genes whose knockout had similar effects on cell proliferation across cell lines). Multiple protein-protein interaction databases and co-expression (correlation in protein abundance across cell lines) in baseline proteomics in Cancer Cell Line Encyclopedia (CCLE; yellow) cell lines were used to provide support for physical or functional interactions among DUBs exhibiting co-dependency. We then used these analyses to explore the impact of DUB knockout on the transcriptome, determine the impact of DUBs on proliferation, and propose E3 ligase interactors and essential functions of DUBs. Acronyms used in figure: knockout (KO), loss of function (LOF), Cancer Cell Line Encyclopedia (CCLE).

DUBs. We then compared drug-induced transcriptomic profiles with those obtained by CRISPR-Cas9-mediated gene knockout.

## The impact of DUB knockouts on the transcriptome

Genes whose knockout with CRISPR-Cas9 resulted in 20 or more differentially expressed (DE) genes in MDAMB231 cells at a false discovery rate (FDR) adjusted p-value of p<0.05 included ten DUBs, one deSUMOylating enzyme (SENP2), and one ubiquitin-like protein (UBL5) (*Figure 2c and d*). To determine if the CMap database, which was collected primarily using a LUMINEX-based 'L1000' method (*Subramanian et al., 2017*), could successfully be queried using 3'DGE-Seq signatures, we focused in on three DUBs that have been relatively well studied: CYLD, TNFAIP3, and PSMD14. Using 3' DGE-Seq data, we found that knockout of PSMD14, a proteosome subunit, strongly perturbed genes involved in the cell cycle (e.g. the *GO cell division* category), as expected for a DUB essential for proteasome function and thus, cell cycle progression (*Figure 2e*, *Supplementary file 2*). When CMap was queried using the 3'DGE-seq data, PSMD14 knockout was found to be similar to knockout of multiple other proteasome subunits (*Supplementary file 3*). In the case of CYLD and TNFAIP3 (also called A20), we found that CRISPR-Cas9 knockout resulted in highly correlated changes in transcription, and GO analysis revealed involvement in NF-κB signaling (*Figure 2d and e*, *Lork et al., 2017*). Both DUBs are known to deubiquitinate members of the NF-κB signaling cascade, such as TRAF2, which results in inhibition of signal transduction (*Lork et al., 2017*). Moreover, we found that 3'DGE-seq data for CYLD and TNFAIP3 knockout were most similar to CMap signatures associated with over-expression of genes that function upstream in the NF-κB pathway, such as the TNF receptor TNFRS1A (*Supplementary file 3*). We interpreted these data on three well-studied DUBs as helping to confirm the validity of our approach.

## Analysis of DUB essentiality using publicly available datasets

Differences in the expression of DUBs in normal tissues and malignant tumors has been described previously (*Luise et al., 2011*), but the direct impact of DUB deletion on specific types of cells has not yet been systematically explored. We therefore investigated the essentiality of DUBs for cancer cells and embryonic development by leveraging three datasets: the DepMap, the International Mouse Phenotyping Consortium (IMPC) dataset, and the Mouse Genome Informatics (MGI) mouse phenotype dataset. Of 94 DUB knockouts found in DepMap, 23 strongly impacted proliferation (dependency score <–0.5) in at least 200 of the cell lines tested (30%), and an additional 25 impacted proliferation in at least eight cell lines (1%) (*Figure 3a*). The remainder had little, if any, detectable effect. To identify dependencies associated with tumor type, we compared DepMap data across sets of cell lines that had been divided based on tissue of tumor origin or more specific clinical or genetic subtypes (e.g. Leukemia is a general category, while AML, ALL, and CML are more specific subdivisions). This can provide insight into disease activity. For example, the tumor type in DepMap most sensitive to knockout of the BRAF kinase is melanoma, the disease in which BRAF inhibitors were first approved and are most widely used (*Kakadia et al., 2018*). We identified 34 DUBs that, when knocked out, significantly and disproportionately affected at least one tumor type more than all other tumor types (two-sided t-test p<0.05, FDR <0.1; *Supplementary file 4*). STAMBP for example, impacts proliferation of head and neck cancer cell lines more strongly (mean dependency score = –0.73) than all other tumor types in the DepMap (mean dependency score = –0.43; difference in means = 0.30 +/- 0.08, FDR = 0.03), suggesting that STAMBP might best be studied in this context; *Figure 3—figure supplement 1* depicts analogous information for other DUBs.

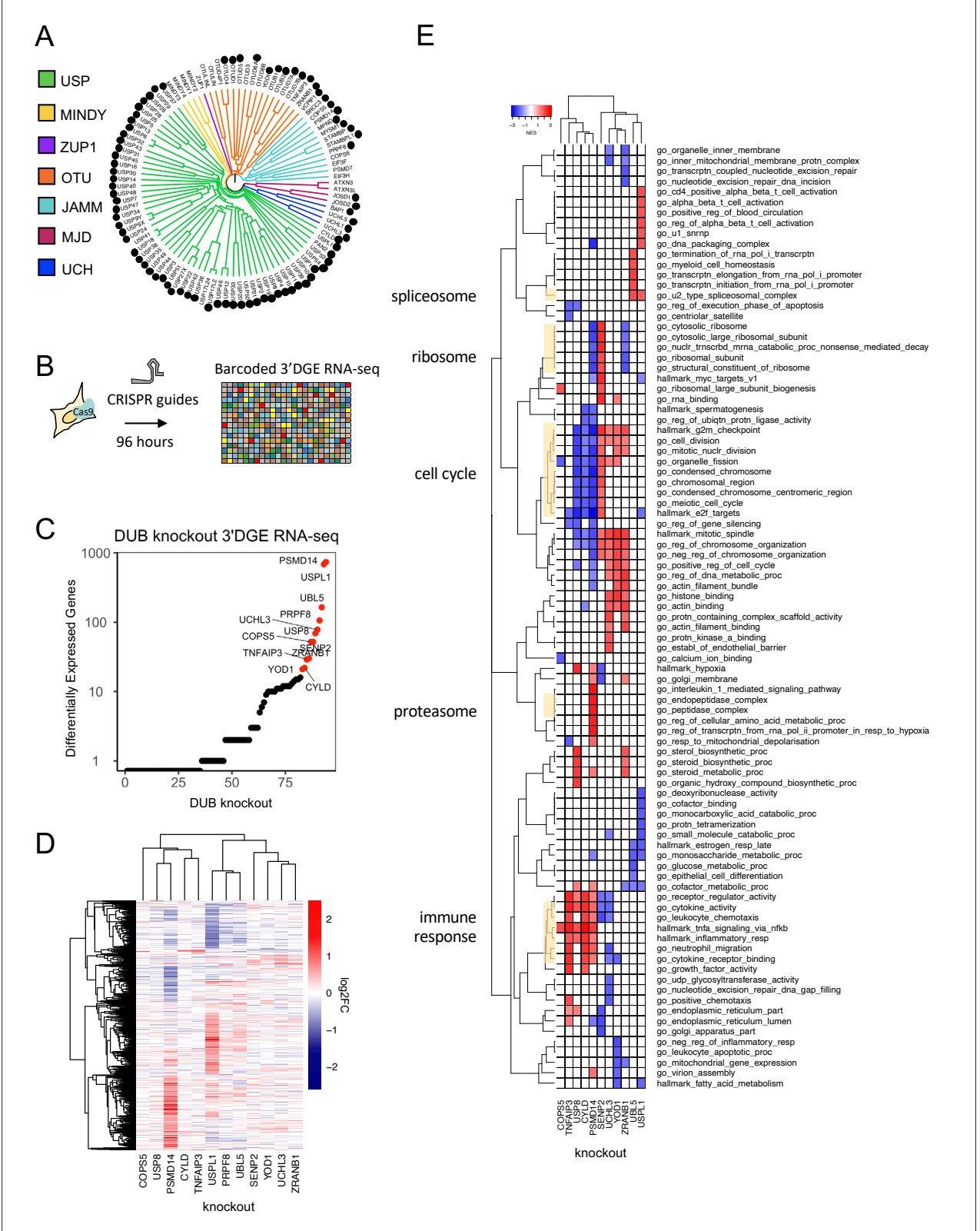

**Figure 2.** Measuring the impact of DUB CRISPR-Cas9 knockout on cell phenotypes. (**A**) 81 DUBs covering the majority of the DUB phylogenetic tree were targeted in MDAMB231 cells using an arrayed CRISPR-Cas9 library (black circles (*Figure 2* cont.) designate knockouts included in screen). (**B**) The impact of DUB loss on the transcriptome was profiled using high-throughput 3' DGE RNA-seq 96 hr after transfection with CRISPR-Cas9. (**C**) The impact of individual DUB knockouts on the transcriptome of MDAMB231 cells, 4 days post CRISPR-Cas9 guide transfection as quantified by the number of

*Figure 2 continued on next page*

*Figure 2 continued*

differentially expressed genes (adjusted p-value <0.05). Guide RNA transfections were performed in triplicate and transcriptional responses then averaged. Knockouts that resulted in more than 20 differentially expressed (DE) genes are colored red. (**D**) Hierarchical clustering of log2FC values for significantly differentially expressed (DE) genes (adjusted p-value <0.05) for the knockouts colored red in (**C**). (**E**) Gene set enrichment analysis results for DUB knockouts that resulted in at least 20 DE genes. Gene sets that were significantly enriched (FDR <0.05) and in the top five up- or down-regulated gene sets in at least one condition are shown. PRPF8 knockout did not result in any significantly enriched pathways so it is not displayed.

The online version of this article includes the following source data and figure supplement(s) for figure 2:

**Figure supplement 1.** Assessment of CRISPR-Cas9 guide target downregulation.

**Figure supplement 1—source data 1.** Original image from western blots in *Figure 2—figure supplement 1* (images for target genes).

**Figure supplement 1—source data 2.** Original image from western blots in *Figure 2—figure supplement 1* (images for loading controls).

**Figure supplement 1—source data 3.** Original image from western blots in *Figure 2—figure supplement 1* (images for target genes, darker exposure).

**Figure supplement 1—source data 4.** Original image from western blots in *Figure 2—figure supplement 1* (images for GAPDH for USP1 KO).

**Figure supplement 1—source data 5.** Original image from western blots in *Figure 2—figure supplement 1* (images for USP1 for USP1 KO).

**Figure supplement 1—source data 6.** Labeled, uncropped blots for *Figure 2—figure supplement 1* (USP7 KO).

**Figure supplement 1—source data 7.** Labeled, uncropped blots for *Figure 2—figure supplement 1* (USP8 KO).

**Figure supplement 1—source data 8.** Labeled, uncropped blots for *Figure 2—figure supplement 1* (USP10 KO).

**Figure supplement 1—source data 9.** Labeled, uncropped blots for *Figure 2—figure supplement 1* (USP1 KO).

**Figure supplement 1—source data 10.** Labeled, uncropped blots for *Figure 2—figure supplement 1* (USP11 KO).

**Figure supplement 1—source data 11.** Labeled, uncropped blots for *Figure 2—figure supplement 1* (UCHL5 KO).

Genes are often studied in a setting or cell type in which they are highly expressed, based on the assumption that level of expression correlates with activity. However, when we compared DUB expression levels and dependency scores, we found that they were uncorrelated (median correlation between DUB dependency score in DepMap and protein abundance in CCLE proteomics data = 0.017, median p-value 0.23), except in the case of TNFAIP3, which was more highly expressed in more sensitive cell lines ($r$=–0.32, p-value =1.6 x $10^{-6}$, *Figure 3—figure supplement 2a*). In fact, a subset of pan-essential DUBs (those DUBs that were essential in >90% of DepMap cell lines) exhibited positive correlation between DUB abundance and DUB dependency score ($r$>0.2), meaning that the most sensitive cell lines had the lowest DUB expression levels (*Ohashi et al., 2019*, *Figure 3—figure supplements 2 and 3*). Thus, with rare exception, the sensitivity of individual tumor lineages to different DUB knockouts is not explained by protein abundance. Moreover, studying DUBs primarily in over-expressing cell lines is not supported by available data.

Using mouse data from the IMPC and MGI datasets, we compared data from cell lines in DepMap to knockout phenotypes in mice (*Figure 3a*, *Muñoz-Fuentes et al., 2018*). Phenotypes in the IMPC are scored prior to weaning in pups that arise from mating heterozygous animals, with lethality at complete penetrance corresponding to no homozygous pups and lethality with incomplete penetrance corresponding to fewer than 12.5% homozygous pups (the expected value for a gene with no impact on survival is 25%). For DUBs with no data in the IMPC, we leveraged knockout mouse data from the MGI dataset, which compiles mouse phenotypes from multiple sources. Of the 82 DUB knockout mice included in the datasets, 20 DUBs were lethal with complete penetrance, 7 were lethal with incomplete penetrance, 47 had non-lethal phenotypes, and 8 resulted in no detectable phenotype in embryos or pups. Of the 27 DUBs that were essential for embryonic development, 21 were also essential for cancer cell viability in at least 1% of cell lines in the DepMap data. A total of 32 DUB knockouts yielded a detectable lethal or non-lethal phenotype in mice but were essential in fewer than 1% of cancer cell lines. Thus, more DUBs were essential in mice than in cell lines – as might have been expected – and the majority of DUBs are likely to have non-redundant functions in development.

## Analysis of co-dependent genes to infer function

To investigate genetic interactions between DUBs and other genes, we performed co-dependency analysis using DepMap data. For each of the 65 DUBs that were essential in ≥3 cells lines, we selected the top seven co-dependent genes and used GO enrichment analysis to identify which protein complexes and pathways were involved (hypergeometric test, FDR-adjusted p<0.05, see Materials

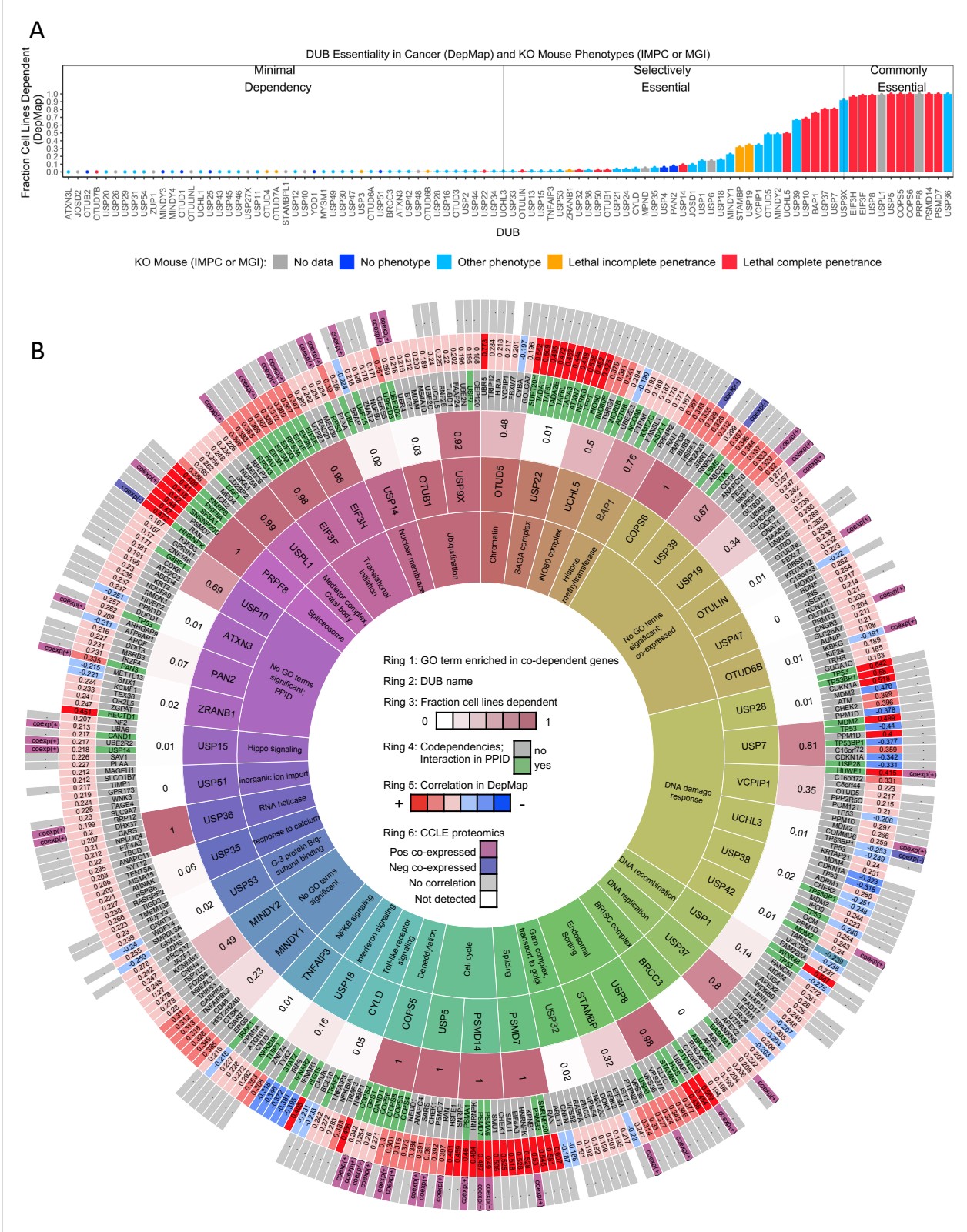

**Figure 3.** DUB essentiality and codependency relationships in DepMap. (**A**) The fraction of cancer cell lines present in the DepMap that are strongly dependent on each DUB (using recommended threshold CERES <–0.5). Bars are colored coded based on knockout mouse phenotype data from the IMPC and MGI datasets. (**B**) The strongest co-dependent genes for each DUB (n=7 but similar results were obtained for n=5–10). For visualization purposes, only DUBs that either had a significantly enriched GO term, a co-dependent gene supported by PPID or CCLE proteomics co-expression,

*Figure 3 continued on next page*

*Figure 3 continued*

or scored as essential in at least 20% of DepMap cell lines are displayed (see *Supplementary files 6 and 7* for complete codependency results). The inner ring (ring 1) contains the top GO term for the co-dependent genes for each DUB (highly similar GO terms are grouped to aid in viewing the data, see *Supplementary file 5* for all GO results). The second ring contains the DUB gene name. The third ring contains the fraction of cell lines strongly dependent on the DUB. The fourth ring contains the co-dependent gene name (green if DUB – co-dependent gene pair exists in a protein-protein interaction database). The fifth ring contains the Pearson correlation value for the DUB-co-dependent gene pair (red represents a positive correlation and blue represents a negative correlation). The sixth ring designates which co-dependent genes had similar transcriptomic profiles in CMap or were co-expressed in baseline proteomics with the respective DUB.

The online version of this article includes the following figure supplement(s) for figure 3:

**Figure supplement 1.** Differential DUB dependency by cancer type.

**Figure supplement 2.** Assessing the association between DUB abundance and dependency.

**Figure supplement 3.** Copy number loss of individual DUBs in various tumor types.

**Figure supplement 4.** Enrichment of interactors and co-expressed genes in co-dependent genes.

**Figure supplement 5.** Enriched GO terms that are common across analyses.

and methods for more detail). Similar results were obtained when the top 5, 7, or 10 co-dependent genes were analyzed by GO enrichment analysis, but setting the threshold at seven genes yielded the largest number of GO terms for well-characterized DUBs that were consistent with literature knowledge. We therefore used a threshold of seven co-dependent genes for the remainder of our analysis. For 35 of 65 DUBs examined, we could identify at least one significantly enriched GO term for co-dependent genes. To enable easy visualization of these and related data, we summarized them as a series of concentric rings (*Figure 3b*); the rings display the most significant GO term (ring 1), the gene names and correlation values of the top seven co-dependent genes (rings 4 and 5), as well as the fraction of cell lines that are dependent on each DUB in the DepMap (ring 3). *Supplementary file 5* provides the same data in tabular form to facilitate computational analysis, and the data are also available on the DUB Portal which is automatically updated weekly to capture public database updates.

For the well-studied DUBs described above, co-dependency analysis returned results consistent with known functions. For example, genes co-dependent with the proteasomal subunit PSMD14 included other members of the proteasome (*GO: endopeptidase complex*), and co-dependent genes for CYLD and TNFAIP3 included members of the NF-κB pathway (*GO: I-κB kinase/NF-κB signaling*) (*Figure 3b*). We therefore asked whether CRISPR-Cas9 knockout of co-dependent genes would elicit similar changes in RNA expression as knockout of the DUB itself. Specifically, for the 9 CRISPR-Cas9 DUB knockouts that resulted in DE of 20 or more genes in MDAMB231 cells, we computed the similarity in CMap for (RNAi-based silencing of) co-dependent genes. We found that four DUBs, CYLD, TNFAIP3, PSMD14, and USP8, had at least one co-dependent gene that, when silenced, resulted in a significantly similar transcriptomic profile to that of the expected DUB knockout (*Supplementary files 6 and 7*). An additional two DUBs, USPL1, and PRPF8, were correlated with splicing factors in both DepMap and CMap, although the specific splicing factors were not the same. Further comparison of DepMap and CMap data was limited by the fact that CMap contains only ~3000 knockdowns as compared to ~18,000 knockouts in the DepMap. We nonetheless conclude that co-dependency analysis yields data on genes that likely interact functionally with DUBs, and the DepMap data and RNA-Seq of CRISPR-Cas9 knockouts were largely consistent in assigning an activity to individual DUBs.

Among the 69 DUBs whose knockout had little or no detectable impact on transcription in MDAMB231 cells (<20 DE genes) in our studies, 55 had little or no impact on proliferation of MDAMB231 cells in DepMap data and 7 were absent from the dataset (*Figure 4—figure supplement 1*). In no case did we detect significant differential gene expression in MDAMB231 cells without evidence of dependency in at least 1% of cell lines (i.e. 8 cell lines). In seven cases however, DUB knockout was associated with a high DepMap dependency score but minimal changes in transcription based on CRISPR-Cas9 screens in MDAMB231 cells. We used western blotting (3 DUBs: USP1, USP7, USP10) or mRNA profiling (1 DUB: USP36) to establish that four of seven target genes in question had actually been downregulated by guide RNA transfection. In these cases, differences in the time and format of the measurement, 4 days after guide RNA transfection for mRNA profiling and 21 days for DepMap data, may explain the difference in transcript profiling and DepMap dependency data. In three other cases (BAP1, USP5, USP37), DUB mRNA was not detectably downregulated in our

MDAMB231 studies, and we assume that the knockout may have failed for technical reasons. Overall, these results suggest good agreement between mRNA profiling and DepMap data with discordance that affected ~5–10% of DUBs (depending on the criterion used) and was potentially explainable by differences in assay format and experimental error.

## Protein interaction and co-expression datasets provide support for co-dependent genes

To search for evidence of physical interaction between genes scored as similar in function to DUBs based on DepMap and CMap data, we mined protein interaction datasets. First, we compiled the protein-protein interactions for each DUB from four PPIDs: BioGRID, IntAct, Pathway Commons, and NURSA (*Cerami et al., 2011*; *Hermjakob et al., 2004*; *Malovannaya et al., 2011*; *Rouillard et al., 2016*). These datasets involve a range of approaches to scoring interaction including affinity capture MS, affinity capture western blotting, and assembly of reconstituted complexes from purified recombinant subunits in vitro. When we asked whether any proteins that exhibited co-dependent genes with DUBs in DepMap data also exhibit interaction with that DUB in PPID data, we found that 31 DUBs (out of 65 DUBs that were essential in ≥3 cell lines in the DepMap), had at least one co-dependent gene that was also an interactor in PPID data (*Figure 3b*, ring 4, interactors shown in green). A total of 55 of 65 DUBs were detectable in CCLE proteomics data (the expression levels of the others were presumably too low) and of those, 24 DUBs were significantly co-expressed with one or more co-dependent genes (FDR <0.01 and |z-score|>2). Moreover, DUBs that are well known to function in multi-protein complexes, such as the USP22 subunit of the SAGA complex, were found to interact with, be co-expressed with, and exhibit co-dependency with other members of the complex. In aggregate, 39 DUBs had at least one co-dependent gene (average 1.6 co-dependent genes per DUB) that was also found to be an interaction partner in a PPID and/or significantly co-expressed in the CCLE baseline proteomics data. Based on these two lines of evidence, we conclude that co-dependent genes for 39 DUBs are likely to interact physically and functionally. These findings are summarized in *Figure 3b* rings 4 and 6 and available in tabular form in *Figure 3—figure supplement 4*. However, some DepMap co-dependent genes were not observed to interact in PPID data, in agreement with the general expectation that proteins functioning in the same pathways might, when perturbed, have a similar effect on cell growth even in the absence of physical interaction.

We performed GO-enrichment analysis on DUB interactors and significantly co-expressed proteins and compared the resulting set of significantly enriched GO terms to the GO terms enriched in the co-dependent genes in the DepMap. This comparison enabled identification of GO terms that are significantly enriched in multiple datasets, providing corroboration for GO terms enriched in the DepMap co-dependent genes. We identified 35 DUBs whose co-dependent genes in the DepMap were associated with one or more significantly enriched GO terms. For 26 of these DUBs, at least one GO term was also enriched in the co-expressed proteins or PPID interactors for the relevant DUB (*Figure 3—figure supplement 5*). Overall, protein co-expression validated more GO complexes than GO pathways, consistent with the idea that the proteome is primarily organized by complexes (*Nusinow et al., 2020*). Moreover, DUBs that have well-characterized functions were often found to be enriched for known interactors or substrates across multiple datasets. For example, the proteasome subunit PSMD14 was enriched for other proteasome subunits and CYLD was enriched for NF-κB signaling in multiple datasets, providing additional confidence in the validity of our approach (*Figure 3—figure supplement 5*).

From these data, we conclude that DepMap co-dependencies, CMap signatures, CCLE proteomic data, and PPIDs provide complementary and consistent data on the likely functions and physical interactors for the great majority of DUBs. These data are summarized in the different rings in *Figure 3*, provided in tabular form in *Supplementary files 6 and 7* are browsable online via the DUB Portal (see methods for full URL). To further increase confidence in results obtained from data mining, we tested specific hypotheses by direct experimentation.

## New insight into the functions of the proteasome-bound DUBs UCHL5 and USP14

Since most well-annotated DUBs have many reported substrates, we sought to use DepMap data to identify which function(s) of DUBs or their substrate(s) might be responsible for cell-essential

phenotypes. For example, UCHL5 is known to interact with both the INO80 complex (which is involved in chromatin remodeling, DNA replication, and DNA repair) and with the proteasome (*Yao et al., 2008*). UCHL5 has been pursued as a cancer therapeutic target because of the latter activity (*D'Arcy et al., 2011*; *Tian et al., 2014*; *Xia et al., 2018*). However, co-dependent gene data from the DepMap show that the effect of UCHL5 knockout is most similar to that of knockout of INO80 subunits (e.g. NFRKB, TFPT, INO80, INO80E, INO80B, r range = 0.29–0.54) (*Figure 3b*). In contrast, no significant correlation was observed with knockouts of proteosome components (e.g. PSMD9, PSMD6, PSMD3, r range = 0.03–0.05). We conclude that UCHL5 is likely to play an essential and non-redundant function not in the proteasome, where it has been most widely studied, but instead in the INO80 complex (*Figure 3b*). This suggests that the therapeutic context for the use of UCHL5 inhibitors is likely to be different from that of proteasome inhibitors, several of which are approved drugs (e.g. bortezomib; *Tan et al., 2019*). More specifically, since the INO80 complex has an essential role in DNA damage repair, (*Yao et al., 2008*) UCHL5 inhibitors may be most useful in combination with DNA damaging agents.

USP14 is another highly studied, proteasome-bound DUB considered to be a promising therapeutic target in some cancers (*Tian et al., 2014*) and USP14 has been reported to rescue many proteins from degradation by the proteasome (*Liu et al., 2018*). We found USP14 to be co-expressed with proteasome subunits in CCLE proteomics data and to interact with the same subunits in PPID data (*Supplementary files 6 and 7*); however, USP14 is not strongly co-dependent with subunits of the proteosome (PSMD13, PSMD7, PSMD4, r range = 0.08–0.09). Instead, a DepMap correlation was observed between USP14 and the UBC polyubiquitin gene ($r = 0.25$), which is one of the primary sources of ubiquitin in mammalian cells (*Figure 3b*). This suggests that loss of USP14 has an anti-proliferative phenotype similar to that of ubiquitin loss and that this phenotype is distinct from that of proteasome inhibition. This hypothesis is consistent with reports that USP14 is required to maintain monoubiquitin pools, and that loss of USP14 leads to an accumulation of polyubiquitin, thereby lowering the levels of free ubiquitin available for conjugation onto protein substrates by E3 ligases (*Lee et al., 2018*; *Yao et al., 2008*). Also consistent with this model are our data showing that CRISPR-Cas9 knockout of USP14 resulted in significant upregulation of the UBC gene but had little additional impact on gene expression (*Figure 4a*). Finally, when we compared the USP14 knockout phenotype to that elicited by exposure of MDAMB231 cells for 24 hr to the USP14 inhibitor I-335 (*Supplementary file 8*), we observed upregulation of UBC and only one other gene (TKT - transketolase - a thiamine-dependent enzyme involved in the pentose phosphate pathway; *Figure 4a*, *Qin et al., 2019*). From these data, we conclude that maintenance of the pool of free ubiquitin, not regulation of the proteasome, is likely to be the key, non-redundant function for USP14.

## USP8 and other ESCRT members impact NF-κB signaling

USP8 is an extensively studied DUB that has been shown to regulate endosomal sorting complexes required for transport (ESCRT). ESCRT complexes recognize ubiquitinated transmembrane receptors and facilitate their transport to lysosomes for degradation (*Mamińska et al., 2016*). USP8 interacts with and stabilizes both receptors and ESCRT proteins. Although USP8 has been reported to regulate the abundance of many proteins (*Dufner and Knobeloch, 2019*), attention has focused on its role in stabilizing EGFR (*Byun et al., 2013*). We found that CRISPR-Cas9 knockout of USP8 also upregulates the expression of cytokines such as IL6 (with a normalized enrichment score – NES of 1.99 and adjusted p-value of 0.005 for '*go_cytokine_activity*') implying that USP8 may have a role in recycling cytokine receptors as well as growth factor receptors. We also observed similarity between the mRNA profiles for USP8 knockout and overexpression of NF-κB signaling proteins such as TNFRSF1A and BCL10 (CMap tau scores: 98.7 and 99.6; *Figure 2e*, *Supplementary file 3*). This is consistent with data showing that knockdown of other members of the ESCRT machinery perturbs cytokine receptor trafficking and results in constitutive NF-κB signaling via TNFRSF1A (the primary TNF receptor; *Mamińska et al., 2016*). We hypothesized that USP8 may impact NF-κB signaling via its role in ESCRT complexes.

To test this hypothesis, we used CRISPR-Cas9 to knock out three ESCRT proteins (UBAP1, HGS, PTPN23) co-dependent with USP8 in DepMap data (*Figure 3b*, *Figure 4—figure supplement 2a*). We found that the transcriptional signature of UBAP1 knockout was strongly correlated with that of USP8 knockout and that both knockouts resulted in upregulation of multiple cytokines (e.g. IL6; *Figure 4b and c*). We also found that, in DepMap data, USP8 knockout exhibited a higher codependency score

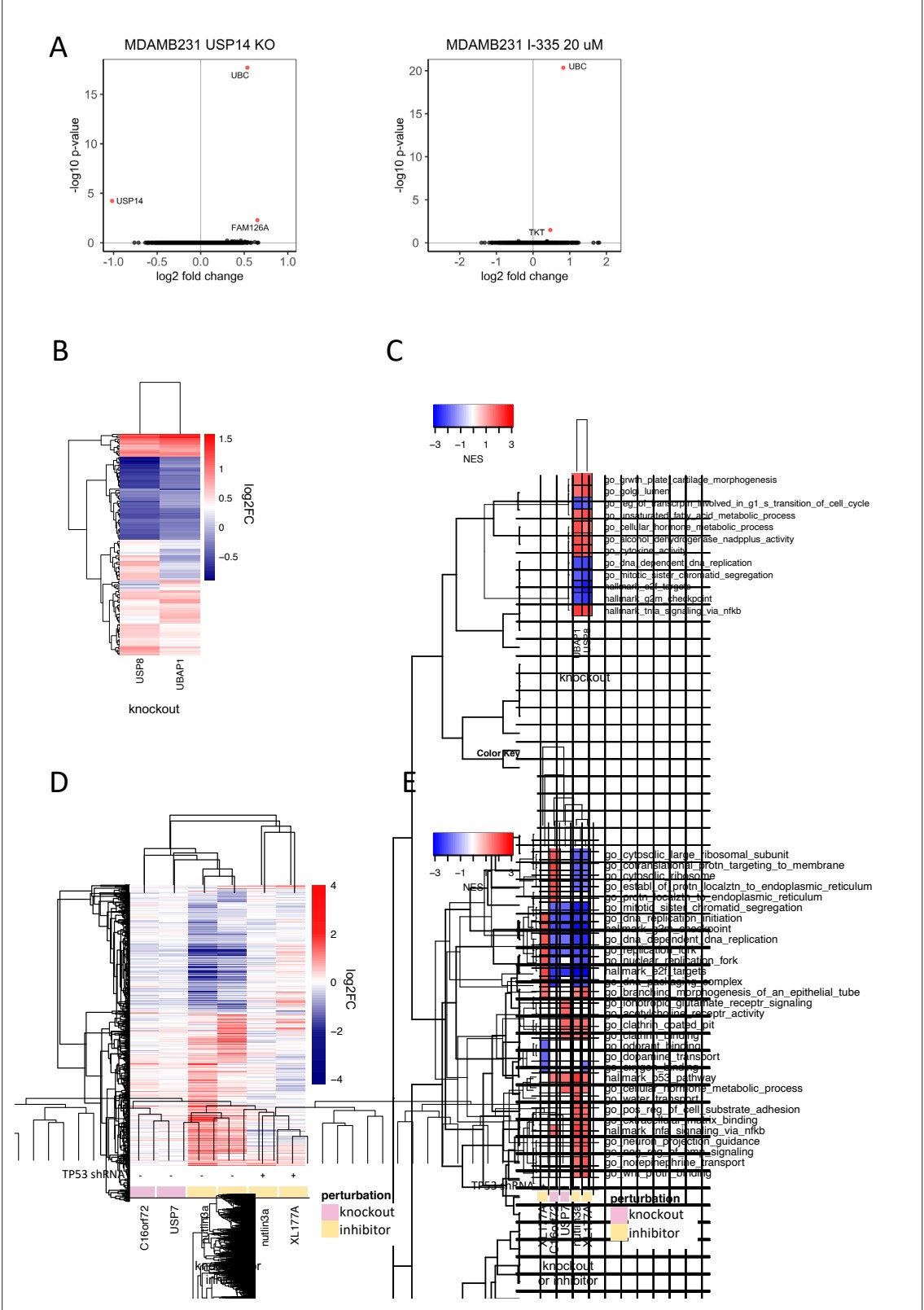

**Figure 4.** Discriminating the functions of well-studied DUBs. (**A**) Changes in gene expression (log2 fold change vs. log 10 adjusted p-value, adjusted p-value <0.05 colored red) in MDAMB231 cells 96 hr following USP14 knockout by CRISPR-Cas9 (left) or 24 hr after treatment with the USP14 inhibitor I-335 at 20 μM (right). (**B**) Hierarchical clustering of significantly differentially expressed genes (adjusted p-value <0.05) 96 hr following knockout of USP8 or UBAP1 in MDAMB231 cells. (**C**) Gene sets significantly (FDR <0.05) enriched in MDAMB231 cells 96 hr after knockout of USP8 or UBAP1. The top

*Figure 4 continued on next page*

Figure 4 continued

five upregulated and top five downregulated gene sets for each condition are shown. (**D**) Hierarchical clustering of significantly differentially expressed genes (adjusted p-value <0.05) 96 hr after knockout of USP7 or C16orf72 in wild-type MCF7 cells or following a 24 hr treatment with 5 µM nutlin3a or 1 µM XL-177A in wildtype or p53 knockdown MCF7 cells. (**E**) Gene sets significantly enriched (FDR <0.05) for the conditions shown in (**D**). The top five up- and down-regulated gene sets for each condition are shown.

The online version of this article includes the following figure supplement(s) for figure 4:

**Figure supplement 1.** Comparison of transcriptional phenotypes and dependency in DepMap.

**Figure supplement 2.** Additional analysis of knockouts of endosomal sorting proteins as well as characterization of small molecules targeting MDM2 and USP7.

(i.e., correlated more strongly) with knockout of ESCRT machinery proteins than with individual growth factor or cytokine receptors, independent of cancer lineage, suggesting that the essential function of USP8 in cancer cells is not mediated by one specific receptor alone – e.g. EGFR – but rather by multiple growth factor and cytokine receptors that undergo similar ESCRT-dependent endosomal sorting.

## USP7 function dependent and independent of functional TP53

USP7 has been the focus of many small molecule inhibitor campaigns (*Kategaya et al., 2017*; *Lamberto et al., 2017*; *Schauer et al., 2020*) and several pharmaceutical companies are developing USP7-based therapeutics, although none have, as yet, advanced to clinical trials. USP7 is reported to regulate chromatin remodeling factors such as polycomb complexes and MDM2, the E3 ligase for the TP53 tumor suppressor protein (*Kim and Sixma, 2017*). We found that MDM2 and other proteins in the TP53 signaling pathway, such as the PPM1D phosphatase, were the strongest co-dependent genes for USP7 (*Figure 3b*). This is consistent with recent work showing that the impact of USP7 on proliferation is strongest in TP53 wild type cell lines (*Schauer et al., 2020*). C16orf72, a protein of unknown function, was another top co-dependent gene for USP7 (DepMap correlation = 0.35; *Figure 3b*). We hypothesized that C16orf72 might be regulated by USP7 and also play a role in the TP53 pathway.

To investigate this hypothesis, we knocked out USP7 and C16orf72 in MCF7 cells using CRISPR-Cas9 and performed 3'DGE-seq after four days. We found that knockout of either gene resulted in a similar mRNA expression profile: in both cases, TP53 pathway genes were upregulated (*Figure 4d and e*). Published proteomic experiments by others show that C16orf72 is one of only eight proteins downregulated two hours after treatment of MM.1S cells with the highly selective USP7 inhibitor XL177A, further suggesting that C16orf72 may be regulated by USP7 (*Bushman et al., 2021*). Our data are also consistent with the sole publication on C16orf72 in PubMed, which describes C16orf72 as a TP53 regulator involved in telomere maintenance (*Benslimane et al., 2021*).

To test directly how much of the USP7 phenotype is dependent on the presence of TP53, we applied 1 µM XL177A (for 24 hr) to isogenic MCF7 cell lines that were WT for TP53 or that had TP53 stably knocked down using shRNA (*Schauer et al., 2020*); we then performed 3'DGE-seq. The MDM2 inhibitor nutlin3a was used as a positive control for TP53 stabilization and activation. MDM2 inhibition with nutlin3a elicited a strong TP53 activation phenotype in parental MCF7 cells (1084 DE genes) but little phenotype in MCF7 TP53 KD cells (only 2 DE genes; *Figure 4d*). Exposure of parental MCF7 cells to XL177A resulted in 737 DE genes, which strongly correlated with DE genes elicited by MDM2 inhibition with nutlin3a as well as knockout of USP7 in parental MCF7 cells (*Figure 4d*). In contrast, exposure of TP53-null cells to XL177A resulted in only 77 DE genes (FDR >0.05) (*Figure 4d*, *Figure 4—figure supplement 2b*). These findings are consistent with the hypothesis that TP53 is a primary target of USP7. The presence of 77 DE genes in XL177A-treated MCF7 TP53 KD cells suggests that USP7 may also have a function independent of TP53 or that XL177A has one or more targets other than USP7 (see below). Overall, these studies nominate a new candidate substrate for USP7 (C16orf72), suggest that C16orf72 plays a role in TP53 signaling, and uncover a possible activity of XL177A that is independent of TP53.

## Additional DUB regulators of TP53

When we looked for evidence that less well-studied DUBs affect TP53 in a manner similar to USP7, we found that VCPIP1, UCHL3, USP38 and USP42 all negatively correlate with TP53 in the DepMap while ATXN3 and USP28 positively correlated with TP53 (*Figure 3b*). The existence of negative correlation in DepMap data is evidence that two genes act in opposing directions on the same pathway. The strongest positive correlation in the DepMap for VCPIP1 was the E3 ligase HUWE1 (correlation = 0.43), suggesting that VCPIP1 might stabilize HUWE1. HUWE1 targets TP53 for degradation via ubiquitination followed by proteolysis by the proteasome, which may explain the negative correlation of VCPIP1 and TP53 in DepMap data (*Figure 3b*). ATXN3 and USP28 have been shown to activate TP53 signaling (*Liu et al., 2016*; *Wang et al., 2018b*), and knockouts of both of these genes positively correlated with TP53 knockout in DepMap data, supporting the hypothesis that these DUBs are positive regulators of TP53. CCLE co-expression and PPID protein-protein interaction analyses also support these findings: VCPIP1 and USP42 are both co-dependent and co-expressed with a negative regulator of TP53 (HUWE1 and PPMID respectively). Additionally, USP38 and USP28 both interact with DNA damage response proteins (*GO response to ionizing radiation*, FDR =7.0 × 10⁻³ and 3.0 × 10⁻³ for USP38 and USP28 respectively), supporting the hypothesis that these DUBs are involved in DNA repair (*Figure 3b*). Overall, these data nominate four DUBs – UCHL3, USP38, VCPIP1, and USP42 – as potential negative regulators of TP53 signaling in addition to the well-established regulator USP7; they also confirm ATXN3 and USP28 as positive regulators.

## New insights into the function of understudied DUBs USPL1 and USP32

Copy number loss of USPL1 is frequent in cancer cell lines and predictive of increased sensitivity to CRISPR-Cas9-mediated knockout of USPL1 in the DepMap (*Figure 3—figure supplements 2 and 3*).

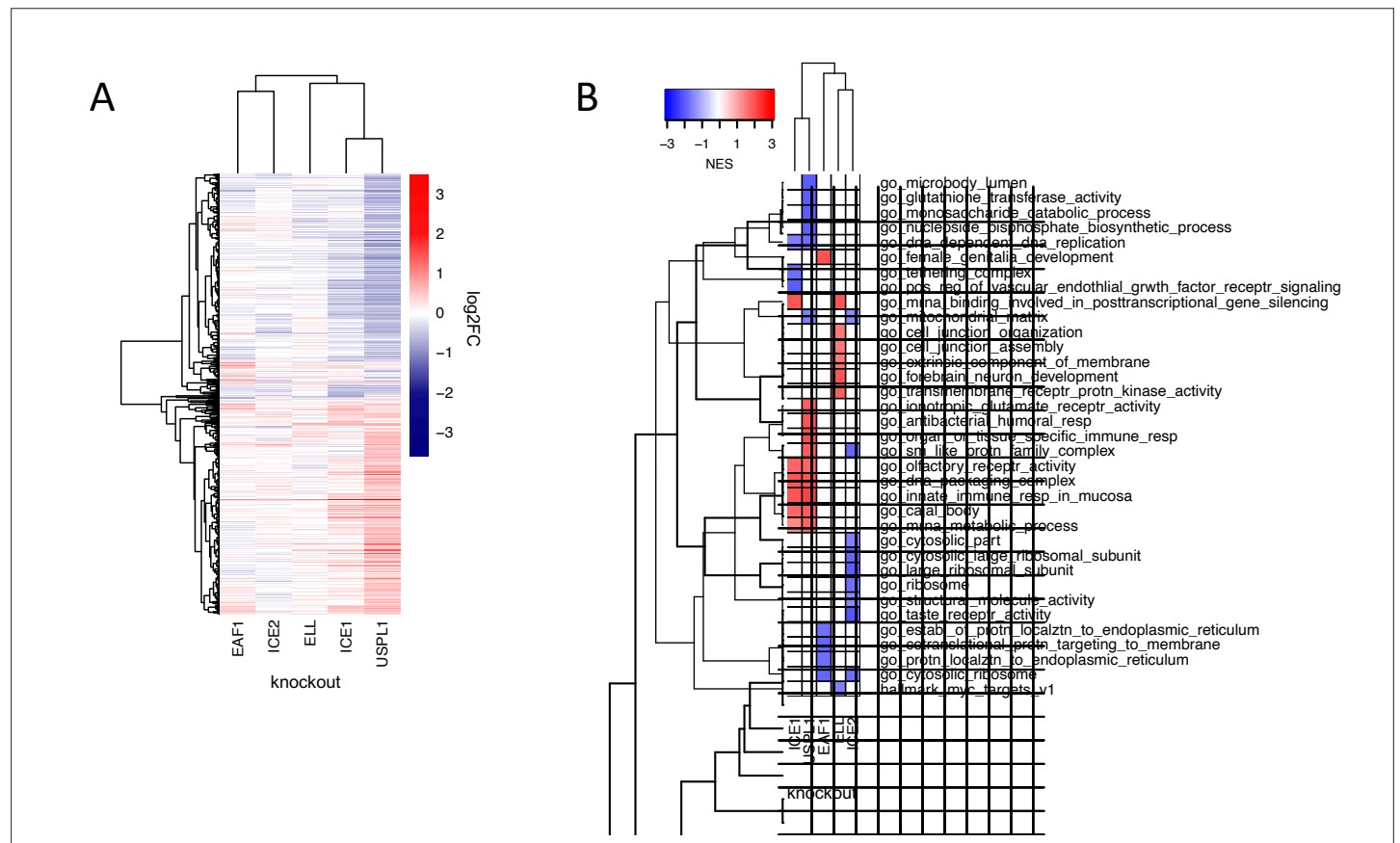

**Figure 5.** The role of USPL1 role in the Little Elongation Complex. (**A**) Hierarchical clustering of log2FC values of differentially expressed genes (adjusted p-value <0.05) 96 h following knockout of USPL1, ICE1, ELL, ICE2, and EAF1 in MDAMB231. (**B**) Gene sets significantly enriched (FDR <0.05) for the conditions shown in (**A**). The top five up- and down-regulated gene sets for each condition are shown.

In our hands, knockout of USPL1 gave the second strongest phenotype in terms of the number of DE genes (*Figure 2c*) and clustered most closely with knockouts of the spliceosome subunit PRPF8 and the 73 amino acid ubiquitin-like protein UBL5, which also plays a role in splicing (*Oka et al., 2014*, *Figure 2d*). Additionally, knockout of USPL1 in MDAMB231 cells resulted in an mRNA expression signature similar to that of knock down of splicing factors such as SNRPD1 in CMap data (*Supplementary file 3*); genes upregulated by USPL1 knockout were enriched in mRNA processing (e.g. *GO: u2 type spliceosomal complex*, FDR *P*-value = 0.03) (*Figure 2e*). The strongest DepMap co-dependent genes for USPL1 across tumor types were members of the Little Elongation Complex (LEC), which is involved in transcription of spliceosomal machinery (*Hutten et al., 2014*) suggestive of an association between USPL1 and the LEC (*Figure 3b*).

To investigate these inferred connections, we used CRISPR-Cas9 to knock out members of the LEC (ICE1, ICE2, ELL, and EAF1) in MDAMB231 cells followed by 3'DGE-seq to score phenotypes. We found that the USPL1 knockout expression signature clustered with signatures for knockout of several of the LEC genes we tested and was most similar to knockout of ICE1 (*Figure 5a*): in both cases, upregulation of genes involved in RNA processing was observed (*Figure 5b*). The high degree of similarity between the USPL1 and ICE1 knockouts is consistent with the DepMap prediction that USPL1 activity is mediated largely by the LEC. USPL1 also interacts with ICE1, ELL, and EAF1 in PPID data, suggesting there is a physical interaction between USPL1 and the LEC. Our findings are also consistent with a previous study showing that USPL1 interacts with subunits of the LEC and affects the localization of spliceosome machinery (*Hutten et al., 2014*; *Schulz et al., 2012*).

USP32 has recently been reported to be important for endosomal sorting to the Golgi apparatus via regulation of the small GTPase, RAB7 (*Sapmaz et al., 2019*). When we ran enrichment analysis on the top USP32 DepMap co-dependent genes, the most significant GO term was *Retrograde Transport Endosome to Golgi.* The co-dependent genes for USP32 include VPS52, VPS54, and RAB6A, which are known to be involved in endosomal sorting to the Golgi apparatus. RAB7 was not a codependent gene, however (correlation = 0.06; *Liewen et al., 2005*). This suggests that the role of USP32 in endosomal transport may be via regulation of the small GTPase RAB6A rather than RAB7 (*Figure 3b*). RAB6 functions in Golgi trafficking, while RAB7 acts more broadly, by associating with late endosomes and lysosomes and regulating diverse trafficking events, including directing late endosomes to the Golgi (*Guerra and Bucci, 2016*; *White et al., 1999*). USP32 is co-expressed with genes involved in retrograde endosome transport to Golgi and the vesicle tethering complex (*GO: Retrograde Endosome Transport to Golgi* and *GO: Tethering Complex*, FDR adjusted enrichment p-values 6.86 × $10^{-6}$ and 1.72 × $10^{-12}$ respectively; *Figure 3b*), providing additional evidence that USP32 is involved in endosomal sorting to the Golgi apparatus.

## Association of DUBs with E3 ligases

When we combined the top seven co-dependent genes for DUBs that impact viability in ≥3 cancer cell lines and ran GO enrichment analysis, we identified strong enrichment for gene sets that included ubiquitin ligases (E3 ligases) and ubiquitin conjugating enzymes (E2 enzymes) (*GO: ubiquitin-like transferase activity*, FDR adjusted p-value =4.9 x $10^{-3}$) (*Figure 7—figure supplement 1*). DUBs are expected to antagonize E3 ligase activity by deubiquitinating E3 ligase substrates, making negative correlations the expected outcome. However, it has also been suggested that DUBs might associate directly with E3 ligases and inhibit their auto-ubiquitination activity, thus preventing proteasomal degradation of the E3 ligase (*Wilkinson, 2009*). In this case, positive correlations in the DepMap between E3 ligases and DUBs are expected. We found that multiple DUBs exhibited strong positive rather than negative correlations with one or more E3 ligases in the DepMap data. We therefore constructed a network of all proteins having ubiquitin transferase activity (including E3 ligases and E2 ubiquitinating proteins) for top co-dependent genes for each DUB (*Figure 6*). This network was found to include many known interactions such as USP7 regulation of the MDM2 E3 and CYLD regulation of the TRAF2 E3. The strongest DUB-ligase correlation in the DepMap was OTUD5 with the UBR5 E3 (*r* = 0.776) which is consistent with previous data that shows that OTUD5 regulates UBR5 (*de Vivo et al., 2019*). Many previously undescribed interactions were also observed, a subset of which are also supported by PPID or co-expression data, including VCPIP1 with HUWE1 and ZRANB1 with HECTD1 (*Figure 6*, circled with dashed lines). These findings suggest roles for 23 DUBs in stabilizing 33 ubiquitin ligases, and provide new insight into which ligases are regulated by which DUBs (*Figure 6*).

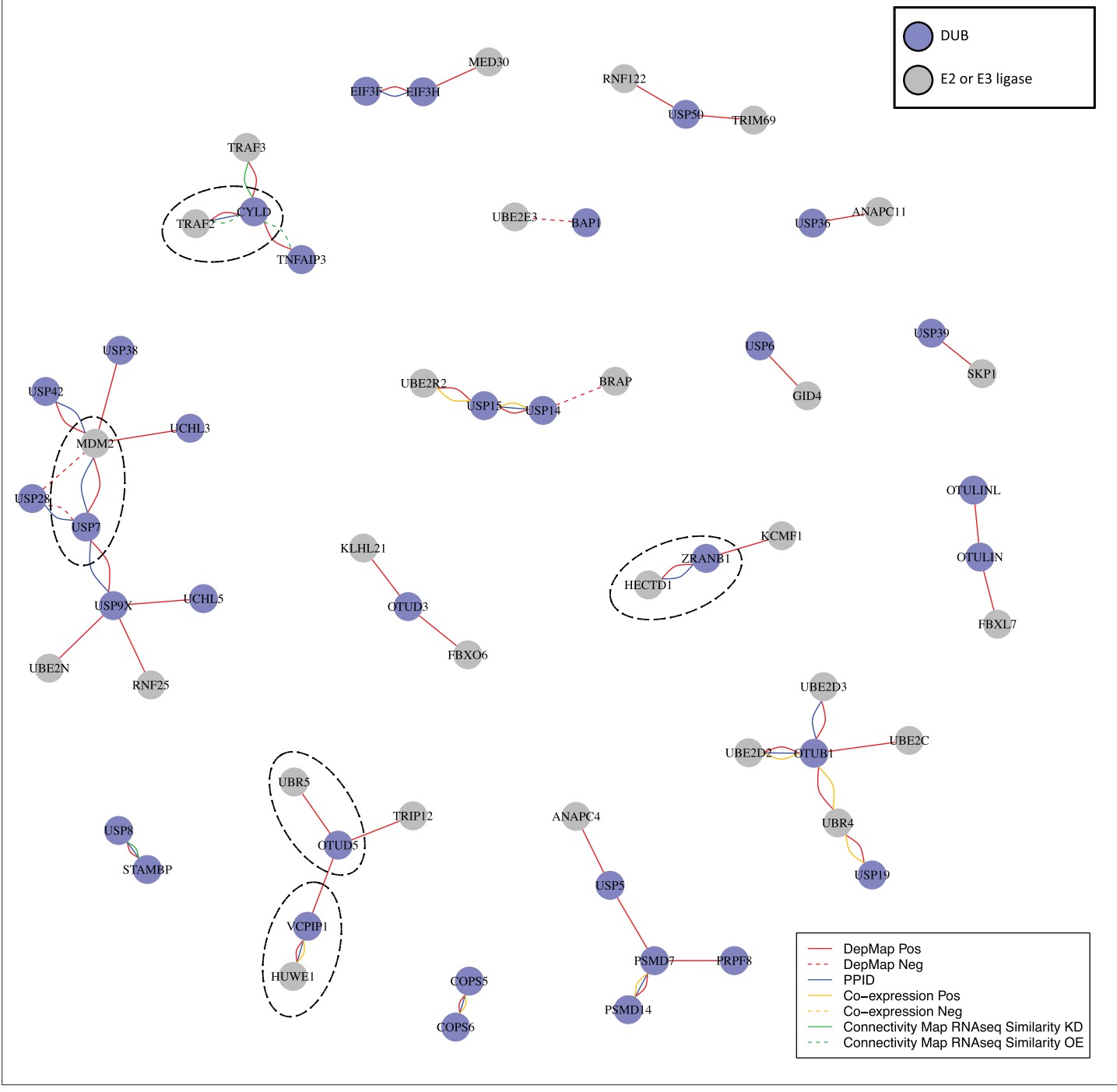

**Figure 6.** DUB E3 ligase network. : Ubiquitin or ubiquitin-like transferases whose co-dependency relationships correlated with DUBs in the DepMap. DUBs are colored blue and ubiquitin transferases are colored grey. Red lines represent correlations in the top seven co-dependent genes. Green lines represent similarity by CMap (tau similarity score >90). Yellow lines represent co-expression in proteomics (FDR <0.01 and |z-score|>2). Blue lines represent interaction in protein-protein interaction databases.

## Comparison of transcriptional impact of small molecule DUB inhibition and DUB knockout

There is growing interest in developing small molecule DUB inhibitors for use as human therapeutics (*Davis and Simeonov, 2015*; *Harrigan et al., 2018*) but the field is still relatively new. We compiled a set of seven recently developed DUB inhibitors that have been described by their developers as being

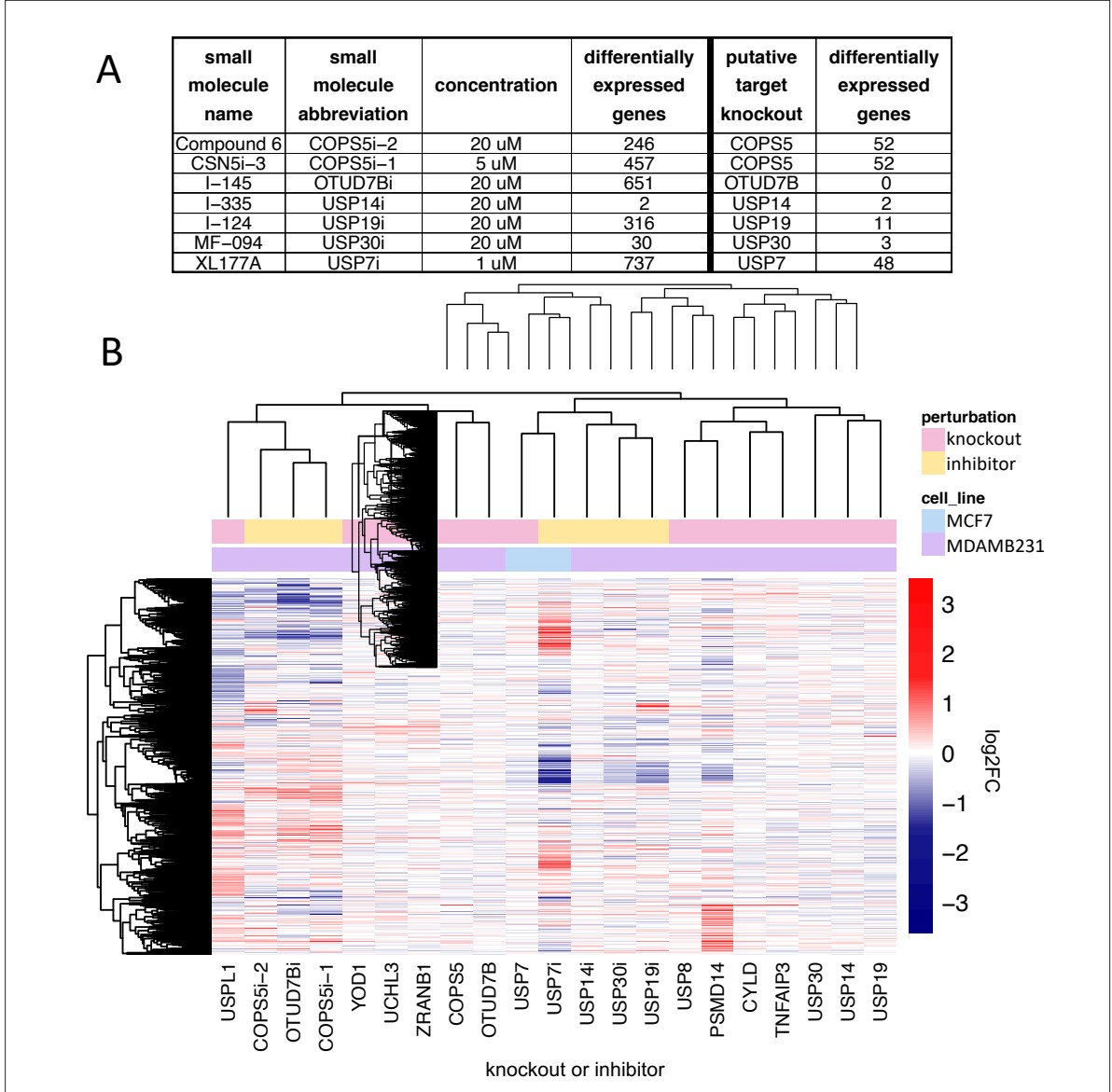

**Figure 7.** Comparison of DUB knockout and inhibition. (**A**) The number of significantly differentially expressed genes (adjusted p-value <0.05) as a result of small molecule DUB inhibition (24 hr treatment) and knockout of the putative target (96 hours after transfection with guide). (**B**) Hierarchical clustering of log2FC values for significantly differentially expressed genes (adjusted p-value <0.05) for small molecule inhibitors of DUBs, the knockout of the putative DUB targets of the small molecules, and the DUB knockout hits that resulted in more than 20 differentially expressed genes. inhibition with XL177A, the DUB inhibitor and knockout were strongly correlated, but the inhibitor resulted in substantially more DE genes, and (iii) in five cases (inhibition of COPS5 with Compound 6 or CSN5i-3, inhibition of OTUD7B with I-145, inhibition of USP19 with I-124, and inhibition of USP30 with MF-094) the inhibitor and knockout had dissimilar signatures (**a** and **b**). CRISPR-Cas9 mediated knockout or small molecule inhibition of USP14 with I-335 resulted in only two DE genes, and the most strongly perturbed DE gene was the same in both instances: UBC. From these data, we conclude that I-335 is a potent and selective inhibitor of USP14.

The online version of this article includes the following figure supplement(s) for figure 7:

**Figure supplement 1.** Gene set enrichment results (Hypergeometric test) of the top seven co-dependent genes for all DUBs using GO Molecular Function gene sets.

selective for one of six DUBs (COPS5, OTUD7B, USP7, USP14, USP19, and USP30); early generation DUB inhibitors were not included since many of these have been shown to have substantial polyphar-macology (*Altmann et al., 2017*; *Kluge et al., 2018*; *Schauer et al., 2020*; *Schlierf et al., 2016*). USP7 inhibitors were studied in the MCF7 breast cancer cell line, which is wild type for TP53 and all other drugs were tested in MDAMB231 cells (which are TP53 mutant). Cells were plated for 24 hr,

exposed to a small molecule for 24 hr at one dose per compound in technical triplicate (*Figure 7a*, *Supplementary file 8*) and mRNA profiling was performed by 3'DGE-seq. We identified significantly perturbed genes (FDR p-value <0.05) for each condition and compared the signatures to those associated with CRISPR-Cas9 knockout of the putative target DUB. We found that treatment of cells with small molecule DUB inhibitors resulted in a median of 10-fold more DE genes than CRISPR-Cas9 knockout of their proposed targets (median DE genes following KO =11, and following inhibitor treatments = 316 *Figure 7a and b*). We then clustered the small molecule 3'DGE-seq signatures with those from our CRISPR-Cas9 DUB knockouts and also used the small molecule signatures to query CMap to identify potential off targets.

Our findings were consistent with three scenarios: (i) in the case of USP14 inhibition by I-335, the mRNA signature was very similar to that of target knockout, (ii) in the case of USP7.

In contrast, inhibition of USP7 with XL177A resulted in 737 DE genes whereas knockout generated 48 DE genes. The two signatures were strongly correlated and querying CMap with the USP7 inhibitor signature also returned USP7 knock down signatures (*Supplementary file 3*). Genes that were significantly DE following XL177A treatment but not USP7 knockout were strongly enriched for TP53 signaling (*Hallmark P53 pathway*, q-value=$3.2 \times 10^{-29}$) and cell cycle pathways (*GO cell cycle*, q-value $1.13 \times 10^{-72}$), which are GO terms also enriched in USP7 knockout DE genes. This suggests that XL177A affects the same TP53 signaling pathway as USP7 knockout but to a greater degree. This difference might reflect differences in time point (24 hr for inhibition vs. 96 hr for knockout), incomplete knockout of USP7 by CRISPR-Cas9, or the existence of additional XL177A targets (some of which could include DUBs other than USP7 shown above to regulate TP53). Not all the effects of XL177A on cells were TP53-mediated however: exposure of TP53 KD cells to XL177A upregulated cell cycle genes, such as genes in the G2 checkpoint (*hallmark G2M checkpoint*, NES =1.94, adjusted p-value =$6.76 \times 10^{-6}$) as well as histone genes (*DNA packaging complex*, NES =1.76, adjusted p-value = 0.0066) (*Figure 4c*).

Exposure of cells to the OTUD7B inhibitor I-145, the USP30 inhibitor MF-094, USP19 inhibitor I-124, or the CSN5 inhibitors Compound 6 or CSN5i-3 resulted in strong perturbation of transcription (651, 30, 316, 246, and 457 DE genes, respectively) whereas CRISPR-Cas9-based knockout of OTUD7B, USP30, USP19, or COPS5 resulted in far fewer DE genes (0, 3, 11, and 52, respectively) and the small molecule and CRISPR-Cas9 signatures were not significantly correlated (*Figure 3a*, *Supplementary file 8*). However, the transcriptomic signature for the OTUD7B inhibitor I-145 had significant similarity to multiple tubulin inhibitors in CMap data (*Supplementary file 3*) but no DepMap dependency was identified for OTUD7B knockout in MDAMB231 cells. The transcriptomic signature of the USP30 inhibitor MF-094 was similar to multiple cyclin dependent kinase inhibitors in the CMap data (*Supplementary file 3*) while the transcriptomic signature of the USP19 inhibitor I-124 was similar in CMap data to the signature associated with overexpression of the cyclin-dependent kinase inhibitors CDKN1A, CDKN2C, or CDKN1B. In CMap data, Compound 6 was dissimilar to COPS5 knockdown (tau =25.4), whereas CSN5i-3 was similar to COPS5 knockdown (tau =94.8). From these data, we conclude that I-145, I-124, Compound 6, and MF-094 (see *Supplementary file 8* for details) are very likely to target proteins other than the DUB they were designed to inhibit. CSN5i-3 may or may not be acting on-target given mixed results (similar by RNAi in CMap but dissimilar to CRISPR-Cas9 knockout) but we cannot rule out inefficient CRISPR-Cas9 knockout or differences in the timing of protein run-down as opposed to inhibition by a small molecule as a contributor to differences in transcriptomic signatures.

## Discussion

Rapid growth in publicly available and 'functional genomic' datasets affords an opportunity for extensive analysis of large gene families such as human DUBs. A total of nine different public resources measuring transcript signatures following gene perturbation (CMap), gene essentiality (DepMap, IMPC, ad MGI), protein co-expression (CCLE proteomics), and protein-protein interaction (BioGRID, IntAct, Pathway Commons PPID, and NURSA PPID) were mined for data, in most cases starting with a CRISPR-Cas9 knockout or small molecule signature we collected in our laboratories. Comparison of enriched gene sets or GO terms made it possible to bridge different types of data. We observed substantial and encouraging consistency among datasets. For example, genes identified as co-dependent with DUBs in DepMap data frequently exhibited similar transcript signatures, were co-expressed

across cell lines, and were physically associated. Overall, our analysis yielded three types of information: (i) potentially new or more precisely specified functions for several well-characterized DUBs, including a UCHL5, USP7, USP8, and USP14 (ii) potential pathways or functional roles for understudied DUBs, including a role for USPL1 in the Little Elongation Complex and UCHL3, USP38, VCPIP1, and USP42 in the regulation of TP53 signaling; (iii) insight into the DUB family as a whole, including evidence that 23 DUBs play a role in stabilizing 33 E3 ubiquitin ligases, most likely by antagonizing their auto-ubiquitination activities. 52 DUBs were found to be essential for proliferation in at least five cancer cell lines, and 34 of these DUBs affected the proliferation of cell lines from one tumor type more than cell lines from all other tumor types, potentially providing insight into disease context. These data are summarized in a simple graphical form in *Figure 3*, as a series of tables suitable for computational analysis in supplementary materials, and in an automatically updating set of tables at the DUB Portal.

## Discriminating among essential and non-essential DUB functions

In several cases, our studies yielded unexpected hypotheses about the functions of DUBs that have already been well studied. For example, USP14 is a component of the proteasome and considered to be a promising therapeutic target in cancer due to the clinical success of other proteasome inhibitors (*Tan et al., 2019*). However, we found that USP14 was strongly co-dependent in DepMap data not with subunits of the proteasome but instead with the UBC polyubiquitin gene, a primary source of ubiquitin in mammalian cells. Knockout of USP14 by CRISPR-Cas9, or exposure of cells to the USP14 inhibitor I-335 resulted in highly selective upregulation of the UBC gene. We therefore propose that maintenance of the pool of free ubiquitin, not regulation of the proteasome, is likely to be the key, non-redundant function for USP14. A similar story emerged for UCHL5, which is a component of both the INO80 complex and the proteasome. The effect of UCHL5 knockout is most similar to that of knockout of other INO80 subunits and no significant correlation was observed in DepMap data with knockout of proteosome subunits. In this case, we hypothesize that UCHL5 plays an essential and non-redundant function in the INO80 complex rather than the proteasome. These data strongly suggest that the therapeutic context for use of USP14 and UCHL5 inhibitors currently in pre-clinical development is likely to be different from that of proteasome inhibitors. More generally, the data show how DepMap data can distinguish among multiple activities for a protein and identify those activities most important for cell survival.

One of the most promising potential uses of DUB inhibitors is to indirectly regulate the levels of disease-associated genes that are not conventionally considered to be druggable such as transcription factors and scaffolding proteins. This strategy has been most actively pursued for USP7, which is a regulator of MDM2, the E3 ligase for the TP53 tumor suppressor protein: inhibition of USP7 increases the levels of ubiquitinated MDM2, promoting its degradation and thereby increasing TP53 levels. Our data on USP7 are consistent with this hypothesis: we find that the top DepMap co-dependent gene for USP7 is MDM2, and knockdown of TP53 largely rescues the transcriptional phenotype observed for USP7 inhibition in TP53 wildtype cells. We find that USP7 has at least one additional substrate, C16orf72, that may also be a TP53 regulator. Moreover, the USP7 inhibitor XL177A has a phenotype that is independent of TP53 and involves upregulation of histone genes and genes involved in the G2M cell cycle checkpoint. We speculate that this may reflect the reported involvement of USP7 in the regulation of polycomb complexes (*de Bie et al., 2010*), although we cannot rule out an off-target activity for XL177A. Regardless, we conclude that the primary role of USP7 in cancer cells involves regulation of the MDM2-TP53 axis (*Schauer et al., 2020*). A number of other DUBs also appear to be TP53 regulators including UCHL3, USP38, VCPIP1, and USP42. Targeting these DUBs in addition to USP7 may be useful as a means to modulate TP53 levels for therapeutic benefit. This could potentially be achieved by developing a small molecule inhibitor active against multiple TP53-regulating DUBs or by using a combination of selective compounds.

## DUBs as E3 regulators

The relationship between USP7 and MDM2 does not appear to be the only instance of a DUB regulating an E3 ligase. DepMap co-dependent genes for DUBs were strongly enriched and positively correlated with E3 ligases and other ubiquitin or ubiquitin-like transferases; in many cases, DepMap data were supported by PPID or co-expression data (e.g. the VCPIP1 DUB and HUWE1 E3 ligase). Overall, we identified 23 DUBs with at least one co-dependent E3 ligase, and eight of these DUBs

had a co-dependent E3 ligase also supported by PPID or co-expression data. Selected DUBs have previously been reported to stabilize E3 ligases; for example, USP7 stabilizes MDM2, CYLD stabilizes TRAF2, and OTUD5 stabilizes UBR5 (*de Vivo et al., 2019*; *Lork et al., 2017*). However, our data suggest that this may be a general feature of the DUB family, with many E3 ligases interacting with DUBs that antagonize E3 auto-ubiquitination and increase protein stability (*Wilkinson, 2009*). Multiple E3 ligases act as oncogenes and promoting their degradation via DUB inhibition may be a broadly useful therapeutic strategy.

## Comparing DUB inhibitors and DUB knockouts

Using transcript profiling we compared seven small molecules reported by their developers to be highly selective inhibitors of specific DUBs to CRISPR-Cas9-mediated knockout of their targets. DUB inhibitor signatures were significantly similar to knockout signatures in only two cases: USP14 inhibition with I-335 and USP7 inhibition with XL177A. We conclude that these compounds are selective, although the signature of USP7 inhibition was substantially stronger than that of USP7 knockout. This was true in general, with exposure of cells to DUB inhibitors resulting, in all cases, in significantly more DE genes than knockouts. In the case of XL177A, our studies cannot determine whether this difference reflects the time at which the measurements were made, the degree of USP7 inhibition by drug or mRNA depletion by CRSPR-Cas9, or the existence of off-target effects. In the cases of the COPS5, OTUD7B, USP19, and USP30 inhibitors, the lack of a significant correlation between weak knockout phenotypes and the strong drug-induced phenotypes suggest substantial off-target activity. Small molecules targeting multi-protein families via competitive inhibition at the active site commonly exhibit some degree of polypharmacology (that is, they exert their biological effects by binding to multiple targets; *Giri et al., 2019*). It appears that, except in the case of USP7 and USP14, additional medicinal chemistry will be required to manage polypharmacology.

It has been suggested that redundancy among DUBs (*Vlasschaert et al., 2017*) might limit the effectiveness of selective DUB inhibitors as therapeutic agents, (*Davis and Simeonov, 2015*). However, we find that single gene knockouts of 43 DUBs impact proliferation in at least 8 DepMap cancer cell lines, and 21 DUBs are embryonic lethal with complete or partial penetrance in mice; deletion of an additional 26 DUBs has a scorable murine phenotype. Thus, many DUBs appear to have non-redundant functions. Moreover, since many targets for successful anticancer drugs are embryonic lethal, (*Yu and Xu, 2020*) our data support further development of DUBs as cancer therapeutics.

## Limitations of this study

The goal of this work was to study the DUB gene family as broadly as possible with limited follow-up on a subset of inferred functions. As a result, deep analysis of individual genes was not possible, and many interferences made from public data are necessarily indirect. We also generate many more hypotheses than we are able to test. The high degree of concordance observed among datasets suggests that pursuing many of these hypotheses will be worth the effort. One nominal 'disagreement' among datasets is essentiality as scored by DepMap data and transcriptional responses as measured by CMap (and our own mRNA profiling studies). For example, seven DUB knockouts impact MDAMB231 viability in the DepMap but do not elicit a strong transcriptional phenotype following CRISPR-Cas9 knockout. We speculate that these discrepancies are due to differences in time point (4 days in the transcriptomics and three weeks in the DepMap screen) or incomplete knockout (although we could confirm successful depletion of four of seven DUBs in question). More study is required to understand the origins and significance of discrepancies between DepMap and CMap. We also found that many physically-interacting and co-expressed genes are not co-dependent in DepMap. Such differences are not unexpected from a biological perspective, given the many different ways in which genes can interact, but further work focused on distinguishing technical errors from functional differences will be important.

Because we analyzed whole gene knockouts rather than point mutations or protein deletions, the results in this paper do not distinguish among catalytic and structural functions for DUBs. This will be an important next step, particularly for clarifying the therapeutic utility of competitive small molecule inhibitors. For example, copy number loss was predictive of increased sensitivity to USPL1 deletion, so we hypothesize that a USPL1 inhibitor may be useful in this context; however, one study suggests

that USPL1 is important for Cajal body architecture independent of its catalytic activity (*Schulz et al., 2012*).

## Conclusion

Our studies provide a diverse set of data on the DUB family as a whole, as well as new insight into many individual DUBs, including several that have been studied intensively. One theme that emerges is that for genes with multiple proposed functions (USP7 and UCHL5 for example), a combination of profiling CRISPR-Cas9 knockouts or drug-induced perturbations with systematic mining of functional genomic databases makes it possible to distinguish among essential and non-essential phenotypes. A second is that more DUBs than anticipated have non-redundant roles in the tumor suppressor and oncogenic pathways, most notably TP53 regulation, suggesting new approaches to undruggable targets. The approaches described in this work are also directly applicable to other gene families and therapeutic targets.

# Materials and methods

**Key resources table**

| Reagent type (species) or resource | Designation | Source or reference | Identifiers | Additional information |
|---|---|---|---|---|
| Genetic reagent (human) | DUB CRISPR-Cas9 screening library: Dharmacon EDIT-R crRNA Library - Human Deubiquitinating Enzymes | Dharmacon (Horizon Discovery) | GC-004700 Lot 17,107 | |
| Genetic reagent (human) | Dharmacon Edit-R tracrRNA | Dharmacon (Horizon Discovery) | U-002005–05 | |
| Transfected construct | Dharmafect 4 | Dharmacon (Horizon Discovery) | T-2004–02 | |
| Cell line (human) | MDAMB231 | ATCC | CRM-HTB-26 | |
| Cell line (human) | MCF7 | ATCC | HTB-22 | |
| Antibody | Flag-tag (L5) antibody (rat monoclonal) | Thermo Fischer | MA1-142 | 1:1,000 dilution |
| Antibody | USP7 antibody (rabbit monoclonal) | Cell Signaling | 4,833 | 1:1,000 dilution |
| Antibody | USP8 antibody (mouse monoclonal) | Santa Cruz Biotechnology | sc-376130 | 1:1,000 dilution |
| Antibody | USP10 antibody (rabbit monoclonal) | Cell Signaling | 8,501 | 1:1,000 dilution |
| Antibody | USP1 antibody (rabbit monoclonal) | Cell Signaling | D37B4 | 1:1,000 dilution |
| Antibody | USP11 antibody (rabbit monoclonal) | abcam | ab109232 | 1:1,000 dilution |
| Antibody | UCHL5 antibody (mouse monoclonal) | Santa Cruz Biotechnology | sc-271002 | 1:1,000 dilution |

## Antibodies, cell lines, and reagents

The Cas9-Flag was a generous gift from Andrew Lane at the Dana Farber Cancer Institute.

## Compounds

All compounds were quality control checked using LCMS and NMR. CSN5i-3 was purchased from MedChemExpress. XL177A, I-335, Compound 6, I-145, I-124, and MF-094 were synthesized according to published methods and compound characterization data matched published data (patents WO2015073528A1, WO2017149313A1, WO2018020242A1) (*Altmann et al., 2017*; *Kluge et al., 2018*; *Schauer et al., 2020*).

## Cell culture

MDAMB231 (ATCC cat no. CRM-HTB-26) was maintained in DMEM media (Corning 10–017-CV) with 10% FBS (Life Technologies 26140–079) and 1% penicillin/streptomycin (Corning 30–002 Cl). MCF7 (ATCC cat no. HTB-22) was maintained in EMEM media with 10% FBS and 1% penicillin/streptomycin. Isogenic MCF7 and MCF7 stable shRNA p53 were a generous gift from the Galit Lahav lab. Cell lines

were maintained in a 5% $CO_2$ incubator at 37 °C, they were identity-validated by STR profiling (**Masters et al., 2001**) and verified to be Mycoplasma-free by the Lonza MycoAlert Kit (Cat. # LT07-318).

Cell lines stably expressing Cas9 were generated by lentiviral infection with pCRISPRV2- FLAG-CAS9 (Addgene #52961) followed by puromycin selection and monoclonal population generation by limiting dilution in 96 well plates. Monoclonal populations with the highest observed knockout efficiency of individual DUBs (USP10 and USP7) were selected.

## CRISPR-Cas9 knockouts and inhibitor treatments for transcriptomic profiling

MDAMB231 Cas9-Flag and MCF7 Cas9-Flag expressing cells were seeded in 96 well plates (4000 cells/well) and allowed to adhere for 24 hr. crRNAs were resuspended in 10 mM Tris-HCl Buffer pH 7.4 (Dharmacon (Horizon Discovery) B-006000–100) and four crRNA guides per DUB were pooled to increase knockout efficiency. Guide transfection was performed according to the recommended manufacturer protocol with optimized conditions as follows. Cells were transfected with 25 nM crRNA and 25 nM tracr RNA using 0.2 µL/well Dharmafect 4, triplicate transfections were performed per condition. The media was replaced 16–18 hr post transfection.

In parallel, MDAMB231 (12,000 cells/well), MCF7 parental (10,000 cells/well), and MCF7 TP53 shRNA (10,000 cells/well) were seeded in 96-well plate format and allowed to adhere for 24 hr. DUB inhibitors and DMSO were dispensed into the 96-well plate using a d300 digital dispenser (Hewlett-Packard).

Cells were lysed 96 hr post crRNA transfection or 24 hr post inhibitor treatment; the plates were washed one time with PBS on a plate washer (BioTek). The PBS was removed (leaving ~15 µL/well residual volume), and 30 µL/well 1 X lysis buffer (1 x Qiagen TCL, 1% beta-mercaptoethanol) was added. The plates were incubated for 5 min at room temperature to aid cell lysis, and then frozen at –80 °C until RNA extraction.

## 3'DGE-seq transcript profiling

The DGE RNA-seq was performed as previously published (**Semrau et al., 2017**; **Soumillon et al., 2014**) with modifications described previously (**Schauer et al., 2020**) (full protocol at https://www.protocols.io/view/3-39-dge-high-throughput-rna-library-preparation-bumynu7w). All automated liquid handling steps described below were performed at the ICCB-Longwood Screening Facility. The cell lysates were mixed and 10 µl was transferred from each well of the 96 well screening plates to a well in a clean 384-well PCR plate, consolidating samples from up to four 96 well plates into a single 384-well plate for RNA extraction. SPRI (solid-phase reversible immobilization) beads, prepared as described previously (**Rohland and Reich, 2012**), were added to the lysate (28 µL/well), mixed, and incubated for 5 min. The beads were then pulled down magnetically, washed with 80% ethanol two times, air dried for one minute, and rehydrated with nuclease free water (20 µL/well). The plate was removed from the magnet, and the beads were resuspended by mixing. After a 5-min incubation, the beads were pulled down again by placing the plate back on the magnet, and the supernatant was transferred to a new 384-well plate. The Qubit Fluorometer and the Agilent BioAnalyzer RNA 6000 Pico Kit were used to verify RNA quantity and quality respectively. RT master mix, 1 µL of barcoded E3V6NEXT adapters, and 5 µl of the total RNA supernatant was transferred to a new 384-well plate for reverse transcription and template switching. All RNA extraction steps were performed with a BRAVO Automated Liquid Handling Platform (Agilent). Following a 90-min incubation at 42 °C, the cDNA was pooled, and the QIAquick PCR purification kit was used for purification. In order to remove excess primers, the cDNA was treated with Exonuclease I for 30 min at 37 °C. The Advantage 2 PCR Enzyme System and the SINGV6 primer were used to amplify the cDNA (5 cycles). Following amplification, Agencourt AMPure XP magnetic beads were used to purify the cDNA and the Qubit Fluorometer was used for quantification. The Nextera DNA kit was used to prepare the sequencing library following the manufacturer's instructions. 55 ng of cDNA was tagmented for 5 min at 55 °C and purified using a Zymo DNA Clean & Concetrator-5 column. The cDNA was amplified (7 cycles) then purified using a 0.9 x ratio of AMPure XP magnetic beads. The Agilent BioAnalyzer HS DNA Kit was used to assess the library size distribution before qPCR quantification and sequencing at the Harvard Medical School Biopolymers Facility (paired end sequencing was performed on an Illumina NextSeq).

The data was separated by well barcode and the reads were converted to counts using the bcbio-nextgen single cell RNA-seq analysis pipeline (https://bcbio-nextgen.readthedocs.io/en/latest/). The pipeline removes any barcodes that differ by more than one base from an expected barcode and uses unique molecular identifiers (UMIs) to identify unique reads and remove PCR duplicates. RapMap was used to align reads to the transcriptome (GRCh38). The R package DESeq2 (version 1.30.0) was used for differential expression analysis, and the R package gseaMultilevel was used for gene set enrichment analysis of all genes sorted by the log2 fold change (adjusted p-value <0.05) using Molecular Signatures Database (MSigDB) gene sets.

To compare the small molecules and CRISPR-Cas9 knockouts, we performed hierarchical clustering of the DE genes of the small molecule treatments, the DE genes of the CRISPR-Cas9 knock outs of each putative target of the small molecules, as well as the DE genes of the DUB CRISPR-Cas9 knock outs that induced the strongest transcriptomic responses in our CRISPR-Cas9 screen (more than 20 DE geness).

## Dependency map analysis

The Broad Institute Dependency Map dataset (CRISPR AVANA dataset 2020 Q3) was analyzed to determine the impact of DUB knockouts on cancer cell lines (*Meyers et al., 2017*; *Tsherniak et al., 2017*). The recommended dependency score threshold of –0.5 was used to score dependent cell lines. The number of cell lines with scores below –0.5 divided by the total number of cell lines tested for a particular DUB was used to determine the fraction of cell lines dependent on a particular DUB. To determine differential response by cancer type, t-tests were conducted to compare the dependency scores for a particular tumor type for a given DUB to the scores of all other cell lines. This was repeated for each tumor type for each DUB, and the p-values were FDR corrected.

To determine co-dependent genes for each DUB, the CRISPR AVANA dataset was used to calculate Pearson correlations between each DUB and all other gene knockouts in the dataset. We limited this analysis to DUBs that had at least three dependent cell lines. To find the pathways and complexes significantly enriched in the strongest co-dependent genes for each DUB, the R package ClusterProfiler was used for gene set overrepresentation analysis of the top 5, 7, or 10 co-dependent genes for each DUB using MSigDB GO gene sets. The overall results were similar, but because the top seven co-dependent gene analysis yielded the largest number of expected GO terms for the well-studied DUBs, the top seven results were used.

In order to extract the associations between DUBs and ubiquitin transferase enzymes, the GO gene set GO Ubiquitin Like Protein Transferase Activity was used to subset DUB co-dependent genes that are ubiquitin or ubiquitin-like transferases.

## Protein-protein interaction database analysis

To compile rich protein-protein interaction data for each DUB, interactions from multiple sources were compiled: IntAct, BioGRID, PathwayCommons, and NURSA.(*Cerami et al., 2011*; *Hermjakob et al., 2004*; *Malovannaya et al., 2011*; *Rouillard et al., 2016*) The R package ClusterProfiler was used for gene set overrepresentation analysis of the interacting proteins for each DUB using MSigDB GO gene sets.

## CCLE proteomics co-expression analysis

The normalized protein abundance data for CCLE cell lines was analyzed to determine genes co-regulated with each DUB(*Nusinow et al., 2020*). Pearson correlations in protein abundance were calculated between each protein in the dataset and each DUB. Only correlations where both proteins were detected in at least 100 cell lines were considered. Significant correlations were selected by thresholding Benjamini-Hochberg adjusted p-values <0.01 (the same significance threshold described in the dataset publication) as well as |z-score|>2. The R package ClusterProfiler was used for gene set overrepresentation analysis of the significant co-expressed genes using MSigDB GO gene sets.

## Overlapping DUB-gene association analysis

To determine which associations between DUBs and genes have support across multiple analyses, the DUB-gene pairs were integrated across the four analyses: top seven DepMap co-dependent genes, CMap (Broad recommended threshold of Score >90), CCLE proteomics coexpression

(Benjamini-Hochberg adjusted p-values <0.01 (the same significance threshold described in the dataset publication) and |z-score|>2), and PPIDs (DUB and gene interact in any of four databases compiled: BioGRID, IntAct, PathwayCommons, or NURSA). Each DUB-gene association was given an evidence count score computed as the sum of the number of analyses that interaction was significant in. Thus, evidence scores range from 0 (DUB gene pair not significant in any of the analyses) to 4 (DUB gene association in all four analyses). The complete table integrating all of these datasets is provided so individual DUBs can be explored (*Supplementary file 6*).

## The DUB portal

To make the data and results presented in this paper available in a reusable form, we also generated online data resources.

First, we created the DUB Portal (RRID:SCR_022476, version 2, https://labsyspharm.github.io/dubportal/), for exploring the most notable results from the experimental and computational analyses for each DUB. The page for each DUB first lists the standard identifier for the related gene, protein, and orthologs in model organisms. It also shows the significantly differentially expressed genes resulting from its knockout in the CRISPR-Cas9 screen described above as well as the significant gene sets calculated by gene set enrichment analysis (GSEA) over the MSigDB.(*Liberzon et al., 2011*) It lists the DUB's top correlations with other genes from the DepMap and provides evidence from PPIDs for direct physical interaction between the correlated genes, when available. In addition, we provide evidence for relations (direct or indirect) between the correlated genes from the INDRA system, which integrates pathway databases and text mined relations from the literature (*Gyori et al., 2017*). The results are further contextualized by presenting the significant Gene Ontology (GO) terms from over-representation analysis. Finally, the portal allows browsing the interactions of each DUB and their supporting evidences collected using INDRA. The DUB portal is automatically generated from source data using Python scripts that standardize the names and identifiers for genes, biological processes, and pathways to promote interoperability (RRID:SCR_022476, version 2, https://github.com/labsyspharm/dubportal, *Hoyt, 2022*).

Second, we added the family- and complex hierarchy of DUB proteins presented in *Figure 2A* to the FamPlex ontology (*Bachman et al., 2018*) and curated cross references to related resources including Medical Subject Headings (MeSH), IntAct, and HGNC Gene Groups (for families of DUB proteins) as well as the Complex Portal and Gene Ontology (for DUB protein complexes). These can be browsed through the FamPlex website at https://sorgerlab.github.io/famplex/ (*Bachman et al., 2018*).

## Acknowledgements

This work was funded by NIH grants U54-CA225088, U54-HL127365 to PKS and R01 CA211681 to SJB, and DARPA grant W911NF2010255 to BMG. This material is based upon work supported by the National Science Foundation Graduate Research Fellowship (NSF grant number: DGE1745303). We thank the ICCB-Longwood Screening Facility for assistance with robotics and liquid handling and Robert Magin and Milka Kostic for valuable discussions on the manuscript.

## Additional information

### Competing interests

Sarah A Boswell: SAB is currently an employee of Ginkgo Bioworks, she declares no conflicts of interest. Sara J Buhrlage: is a member of the SAB of Adenoid Cystic Carcinoma Foundation. In the last five years the Buhrlage lab has received research funding from AbbVie and in-kind resources from Novartis Institutes for Biomedical Research. Buhrlage declares that none of these relationships have influenced the content of this manuscript. Peter K Sorger: PKS is a member of the SAB or BOD member of Applied Biomath, RareCyte Inc., and Glencoe Software; PKS is also a member of the NanoString and Montain Health SABs. In the last five years the Sorger lab has received research funding from Novartis and Merck. Sorger declares that none of these relationships have influenced the content of this manuscript. The other authors declare that no competing interests exist.

## Funding

| Funder | Grant reference number | Author |
|---|---|---|
| National Institutes of Health | U54-CA225088 | Peter K Sorger |
| National Institutes of Health | CA211681 | Sara J Buhrlage |
| Defense Advanced Research Projects Agency | W911NF2010255 | Benjamin Gyori |
| National Science Foundation | DGE1745303 | Laura M Doherty |
| National Institutes of Health | U54-HL127365 | Peter K Sorger |

The funders had no role in study design, data collection and interpretation, or the decision to submit the work for publication.

## Author contributions

Laura M Doherty, Conceptualization, Data curation, Formal analysis, Investigation, Methodology, Software, Supervision, Validation, Visualization, Writing – original draft, Writing – review and editing; Caitlin E Mills, CEM provided valuable scientific feedback., Writing – original draft, Writing – review and editing; Sarah A Boswell, Investigation, SAB did the RNA extraction and prepared the library for RNA-sequencing, Writing – review and editing; Xiaoxi Liu, Investigation, Writing – review and editing, XL synthesized the small molecule inhibitors; Charles Tapley Hoyt, Benjamin Gyori, CTH and BMG built the DUB Portal, Software, Writing – review and editing; Sara J Buhrlage, Conceptualization, Funding acquisition, Supervision, Writing – original draft, Writing – review and editing; Peter K Sorger, Conceptualization, Supervision, Writing – original draft, Writing – review and editing

## Author ORCIDs

Laura M Doherty (b) http://orcid.org/0000-0002-8171-5109
Caitlin E Mills (b) http://orcid.org/0000-0002-2608-4084
Sarah A Boswell (b) http://orcid.org/0000-0002-3118-3378
Charles Tapley Hoyt (b) http://orcid.org/0000-0003-4423-4370
Benjamin Gyori (b) http://orcid.org/0000-0001-9439-5346
Sara J Buhrlage (b) http://orcid.org/0000-0003-4562-1823
Peter K Sorger (b) http://orcid.org/0000-0002-3364-1838

## Decision letter and Author response

Decision letter https://doi.org/10.7554/eLife.72879.sa1
Author response https://doi.org/10.7554/eLife.72879.sa2

# Additional files

## Supplementary files

- Supplementary file 1. cRNAs used for CRISPR-Cas9 knockout studies.

- Supplementary file 2. Full results for gene set enrichment analysis for CRISPR-Cas9 knockout studies.

- Supplementary file 3. Full results for Connectivity Map analysis for CRISPR-Cas9 knockout studies.

- Supplementary file 4. Full t-test results for lineage sensitivity analysis of DUBs in the DepMap.

- Supplementary file 5. Dataset integration results used to generate *Figure 3*.

- Supplementary file 6. Full dataset integration results for integration of DepMap, CMap results, CCLE proteomics, and PPIDs.

- Supplementary file 7. Column descriptors to facilitate interpretation of *Supplementary file 6*.

- Supplementary file 8. Identifiers for small molecules used in RNA-seq screen.

- Transparent reporting form

## Data availability

All raw RNAseq data and differential expression results are available on synapse (syn25008205). Additionally, all RNAseq data are available on GEO (accession number: GSE187008). All other data pertaining to the RNA-seq studies are included as Supplemental Tables: Supplementary File 1: cRNAs used for CRISPR-Cas9 knockout studies. Supplementary File 2: Full results for gene set enrichment analysis for CRISPR-Cas9 knockout studies. Supplementary File 3: Full results for Connectivity Map analysis for CRISPR-Cas9 knockout studies. Figure 2-figure supplement 1 source data 1-11: Original images from western blots in Figure 2-figure supplement 1. Supplementary File 8: Identifiers for small molecules used in RNA-seq screen. Data from integration of datasets is available in supplemental tables: Supplementary File 4: Full t-test results for lineage sensitivity analysis of DUBs in the DepMap. Supplementary File 5: Dataset integration results used to generate Figure 3. Supplementary File 6: Full dataset integration results for integration of DepMap, CMap results, CCLE proteomics, and PPIDs. Supplementary File 7: Column descriptors to facilitate interpretation of Supplementary File 6.

The following datasets were generated:

| Author(s) | Year | Dataset title | Dataset URL | Database and Identifier |
|---|---|---|---|---|
| Doherty LM, Mills CE, Boswell SA, Liu X, Hoyt CT, Gyori BM, Buhrlage SJ, Sorger PK | 2019 | DUB CRISPR RNAseq screen | https://www.synapse.org/#!Synapse:syn25008209 | Synapse, syn25008209 |
| Doherty LM, Mills CE, Boswell SA, Liu X, Hoyt CT, Gyori BM, Buhrlage SJ, Sorger PK | 2020 | DUB inhibitors and targeted studies RNAseq | https://www.synapse.org/#!Synapse:syn25008216 | Synapse, syn25008216 |

The following previously published dataset was used:

| Author(s) | Year | Dataset title | Dataset URL | Database and Identifier |
|---|---|---|---|---|
| Malovannaya A, Lanz RB, Jung SY, Bulynko Y, Le NT, Chan DW, Ding C, Shi Y, Yucer N, Krenciute G, Kim B-J, Li C, Chen R, Li W, Wang Y, O'Malley BW | 2011 | NURSA Protein Complexes | https://maayanlab.cloud/Harmonizome/dataset/NURSA+Protein+Complexes | NURSA, Protein Complexes |
| Nusinow DP, Szpyt J, Ghandi M, Rose CM, Robert McDonald E, Kalocsay M, Jané-Valbuena J, Gelfand E, Schweppe DK, Jedrychowski M, Golji J, Porter DA, Rejtar T, Karen Wang Y, Kryukov GV, Stegmeier F, Erickson BK, Garraway LA, Sellers WR, Gygi SP | 2020 | CCLE proteomics | https://depmap.org/portal/download/?release=Proteomics | DepMap portal, CCLE proteomics |

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
