## [Editor Report]

This study reports the creation of a database on deubiquitinating enzymes (DUBs), which integrates existing large-scale datasets with new knock-out and inhibition experiments. The combined data confirm known DUB functions and, importantly, correct several current assumptions and highlight potential new functions of DUBs. The data are made available through an online portal, providing a useful resource for investigators interested in DUB functions or considering DUBs as drug targets.

---

## [Decision Letter]

**Decision letter after peer review:**

Thank you for submitting your article "Integrating multi-omics data reveals function and therapeutic potential of deubiquitinating enzymes" for consideration as a Tools and Resources article by *eLife*. Your article has been reviewed by 3 peer reviewers, and the evaluation has been overseen by a Reviewing Editor and Volker Dötsch as the Senior Editor. The following individual involved in review of your submission has agreed to reveal their identity: Benedikt M Kessler (Reviewer #3).

The compilation of all the available DUB information in one resource was seen as a useful contribution to the field by the majority of the reviewers. To be acceptable for *eLife*, a better presentation of the data (including in the online portal), as well as clearer explanations on steps taken and thresholds applied will be needed.

Essential revisions:

1. A major question that arose was how quickly the DUB portal is going to be outdated. Surely, for example, the number of papers column will be outdated in the near future. Are the underlying databases and the statistical analysis linked to the portal in a way that any new entries to these databases or any new relevant large datasets will be reflected in the DUB portal? This would make it a reliable resource for the community in the long term.

2. Better presentation of the data:

– The "synthesis" of data that the paper promises, and which to some extent is covered in the paper in Figure 3, is not well reflected in the online portal. Could there be a more innovative way, rather than tables, to present the data? A real benefit of this paper is seeing correlation or consistency among multiple data sources – so this should be better highlighted.

– Columns in the online portal should be explained and defined within the portal.

– The "About" page in the DUB Portal seems undeveloped.

– Figure 1 was experienced as confusing by several reviewers. The figure implies that data is entered into the databases, when mostly data is taken out of databases. Please make it match with your workflow in a more intuitive way. This figure is important for readers to be able to follow the strategies taken.

– For a reader of the paper, it would be interesting to know to what extent the new data (knockout/RNA-seq of 81 DUBs) has changed the conclusions or enabled conclusions that were not possible with only the mining of the existing databases.

– Figure 3B: USP5 is represented twice, and one of these instances should probably instead be USP53.

– Please provide the GEO accession number for the new data.

– The abstract could better reflect the findings. A current focus is on c-myc, where it is somewhat unclear what new biology has been discovered.

– In the discussion, page 30, "… by a molecule that inhibits multiple TP53-regulating DUBs". Do multiple TP53-regulating DUBs have highly similar binding pockets? And wouldn't such nonspecific inhibitors have more off-target effects? Is there prior evidence that makes this a viable strategy?

3. Clearer explanations on steps taken and thresholds applied:

– Knockout/RNA-seq data and Figure 2A: Why were those 81 DUBs chosen?

– Many terms used are highly specialized and not easily accessible to the general reader of *eLife*. The manuscript would benefit from an effort to provide simplified explanations for the rationale and what can be extrapolated in biological terms.

– Please provide information on how thresholds were selected. For example, genes whose knockout resulted in 20 or more DE genes – why 20? Page 24: why top 7 co-dependent genes? (Sometimes, details are available in the methods section, but it would help readers to have it briefly mentioned in the main text as well.) What do the authors consider a significant "normalized enrichment score", what a good "CMap tau score"?

4. Since your paper is considered for the "Tools and Resources" section, we won't make this a prerequisite for publication, but the confidence of the community in the mined data would be greatly enhanced by providing at least one targeted follow-up experiment on one of the novel hypotheses generated – for example substantiating novel findings presented in Figure 6.

---

## [Author Response]

Essential revisions:1. A major question that arose was how quickly the DUB portal is going to be outdated. Surely, for example, the number of papers column will be outdated in the near future. Are the underlying databases and the statistical analysis linked to the portal in a way that any new entries to these databases or any new relevant large datasets will be reflected in the DUB portal? This would make it a reliable resource for the community in the long term.

In response to this concern we have implemented a new, automated workflow that updates the information in the DUB portal on a regular basis. We are grateful for this suggestion and feel that the updates will help to make it a useful resource in the medium to long term. We describe below the specific elements of the website that have now been automated.

The DUB Portal website is hosted on GitHub and therefore has version control. We used GitHub Actions to create a continuous integration service that automatically rebuilds the DUB Portal on a weekly basis. The build date and the versions of the resources used for the current DUB Portal are now shown on each page. Each build has an associated commit in version control, so it is always possible to retrieve and re-deploy an old version of the DUB Portal if desired. Each build of the DUB Portal now includes information derived from the following actions:

– Querying PubMed for the number of publications associated with a given DUB gene. As the reviewers pointed out, this is almost certain to increase over time.

– Querying the INDRA Database for the latest statements associated with a given DUB gene (the INDRA Database itself is updated nightly). INDRA includes the protein-protein interaction databases, so new entries to these databases will get added to the DUB portal. These databases are always expanding, so this automation will also add data moving forward.

– Querying Reactome for the latest pathway annotations.

– Retrieving explanations for DUB dependencies derived from integrated analysis of BioGRID, IntAct, PathwayCommons, has also been automated. Data from BioGRID, IntAct, and Pathway Commons are re-queried on a weekly basis. NURSA is a static data source (i.e., we obtained it from the publication) so therefore it does not update and does not need to be repeatedly queried, Furthermore, BioGRID and PathwayCommons are included within INDRA, so new additions to these datasets will be included in the Evidence column of the appropriate DUB Portal table along with a link to a page that explains which datasets provided a specific piece of INDRA evidence. We describe these new features of the DUB Portal in an updated version of the *About* page.

As described above, we have also integrated the graphics of the manuscript with the tables in the Portal to make access to the underlying information more intuitive.

2. Better presentation of the data:– The "synthesis" of data that the paper promises, and which to some extent is covered in the paper in Figure 3, is not well reflected in the online portal. Could there be a more innovative way, rather than tables, to present the data? A real benefit of this paper is seeing correlation or consistency among multiple data sources – so this should be better highlighted.

In response to this suggestion, we have made numerous changes to the DUB Portal and the way it generates data. However, for the Portal to be updatable, it requires that data we read into the portal have a structured format such as tables with hyperlinks. This is not easily converted to the more visually appealing but inherently unstructured, static figures we have created for the publication itself. While an expert in data visualization might be able to program a site in which tables are used to automatically generate a new version of Figure 3 (the circular plot), this is not a function found in tools such as R-Shiny. Such a complex task seems to us beyond the scope of the current paper and our own expertise in data visualization.

We note that even if the tabular presentation of data in the DUB Portal is less visually striking, it includes additional information that is not included in the manuscript itself such as an *evidence* column that provides information on correlation or consistency among multiple data sources. This is an important type of information for integrating data across types and sources.

(As mentioned above – we have also added Figure 3 to the website to capture the integration in a visually appealing way and better key the summary figure to the tabular information).

– Columns in the online portal should be explained and defined within the portal.

This is an excellent suggestion. Each header field in the tables now has hover text explaining its meaning. A more detailed explanation for each field is provided on the *About* page.

– The "About" page in the DUB Portal seems undeveloped.

We have significantly extended the *About* page to provide detailed documentation of the DUB Portal and each of its components. The About page also contains metadata on the version of different resources used to build the portal and links to Github where issues can be submitted.

– Figure 1 was experienced as confusing by several reviewers. The figure implies that data is entered into the databases, when mostly data is taken out of databases. Please make it match with your workflow in a more intuitive way. This figure is important for readers to be able to follow the strategies taken.

We have generated an updated version of Figure 1 that more closely matches the actual workflows. In this revised figure datasets are placed above one arrow that conveys their use in achieving the results below the arrow. We hope that the reviewers find this representation more intuitive. If not, we are happy to consider additional changes.

– For a reader of the paper, it would be interesting to know to what extent the new data (knockout/RNA-seq of 81 DUBs) has changed the conclusions or enabled conclusions that were not possible with only the mining of the existing databases.

This is an interesting point that cannot be answered precisely since it is a counterfactual scenario. However, we found that collecting a set of original (experimental) data on DUB knockouts was essential for us to gain traction with online resources, particularly the Connectivity Map. To query the Connectivity Map, an RNA-seq signature is required. There are not many DUB knockdowns in the Connectivity Map, and thus, we could not query the gene family directly. Moreover, signatures in the Connectivity Map were collected using an unusual L1000 Luminex bead-based assay that does not allow for conventional gene set enrichment. Thus, with respect to DUB biology the transcriptional knockout signatures we have collected are potentially more broadly useful than those in the Connectivity Map. One limitation in our study is that we did not use a CRISPR-Cas9 library for all 94 DUBs that were subsequently included in our bioinformatic analysis; however, as mentioned above, Dharmacon is interested in using data in the current paper to enhance its library.

We now highlight a specific use case of newly acquired RNA-seq dataset in the *Discussion*: this involves a comparison of CRISPR knockout RNA-seq signatures with signatures for small molecule inhibitors. Such an analysis would not have been possible without newly acquired data since no database of which we are aware has the necessary information on CRISPR and small molecule perturbations.

Moreover, in the section of the results describing USP8 we discuss use of our RNA-seq data and the DepMap to generate hypotheses about USP8 biology. The RNA-seq data gives insight into the downstream pathways that are perturbed (e.g. cytokine expression in the case of USP8 knockout) which provided additional insight beyond the functions predicted by the databases (endosomal sorting in the case of USP8). Many receptors are trafficked through endosomes, so predicting that cytokine signaling would be the dominant pathway perturbed by loss of ESCRT machinery would not have been intuitive.

– Figure 3B: USP5 is represented twice, and one of these instances should probably instead be USP53.

Thank you for catching this typographical error. We updated Figure 3B to correct the instance of USP5 that is supposed to be USP53.

– Please provide the GEO accession number for the new data.

The GEO accession number (GSE187008) has been added to the Data Availability section.

– The abstract could better reflect the findings. A current focus is on c-myc, where it is somewhat unclear what new biology has been discovered.

Thank you for this suggestion. We removed the reference to c-myc and instead highlighted the individual DUBs that are the focus of the follow up studies in the paper.

– In the discussion, page 30, "… by a molecule that inhibits multiple TP53-regulating DUBs". Do multiple TP53-regulating DUBs have highly similar binding pockets? And wouldn't such nonspecific inhibitors have more off-target effects? Is there prior evidence that makes this a viable strategy?

Thank you for highlighting this issue. Our study suggests it would be worth targeting the redundancy of the DUB p53 regulatory mechanism. However, we were actually imagining that it might be necessary to develop a combination therapy to achieve this. The possibility that a multi-targeting compound might be developed is less certain, for the reasons the reviewer describes. To make it clear that multiple, selective DUB inhibitors may be the easiest way to address redundancy we modified the sentence in the Discussion to:

“This could potentially be achieved by developing a small molecule inhibitor active against multiple TP53-regulating DUBs or by using a combination of selective compounds.”

3. Clearer explanations on steps taken and thresholds applied:– Knockout/RNA-seq data and Figure 2A: Why were those 81 DUBs chosen?

At the time this part of our study was performed, it seemed logical to use a commercially available guide RNA collection, in part because it would then be easy for others to repeat relevant aspects of our study. We therefore used a library from Dharmacon that is predominantly focused on DUBs. To our knowledge, there is no specific reason that the 81 DUBs in this library were selected out of ~100 in the whole genome. One factor is that, at the time that the library was constructed, annotation of DUBs was incomplete. For example, there are no guides in the library targeting the genes in the recently discovered MINDY and ZUP1 DUB families. To make this point clear we have modified the main text to read:

“To generate knockouts, we used a commercially available arrayed CRISPR-Cas9 library targeting 81 out of ~100 DUBs and 13 additional proteins in the ubiquitin-proteasome system, including ubiquitin-like proteins…”

We have written to Dharmacon to recommend that they update the collection in the DUB focused CRISPR library to make it more comprehensive and up to date, and they stated they are now considering this possibility. We anticipate that publication of this manuscript will increase their enthusiasm.

– Many terms used are highly specialized and not easily accessible to the general reader of eLife. The manuscript would benefit from an effort to provide simplified explanations for the rationale and what can be extrapolated in biological terms.

The reviewer makes an excellent point! In response, we have made two changes to the manuscript. First, we have added a new Table (Table 1) that briefly describes specific data types as well as their key features, parameters, and abbreviations. Second, we have revised the text to ensure that a rationale is included for each of the datasets collected and resources used. This rationale is included in the first section of the results so the reader will understand the approach before detailed information on individual DUBs is presented.

For example, we expanded the rationale for collecting knockout RNA-seq data and using the CMap, which now states:

“As a first step, we sought to leverage CMap to identify genes that, when silenced with RNAi, or overexpressed, had similar transcriptional effects as DUB knockouts. Perturbations that act on the same pathway often share similar transcriptomic signatures. (Lamb et al., 2006)”

The rationale for the DepMap analysis is:

“It has been observed that genes with similar DepMap scores across cell lines are more likely to have related biological functions (Meyers et al., 2017; Pan et al., 2018; Tsherniak et al., 2017), a property known as co-dependency. More specifically, co-dependent genes are frequently found to lie in the same or parallel pathways (as defined by gene ontology (GO), for example) or to be members of the same protein complex. We identified co-dependent genes for DUBs and then ran GO enrichment analysis to identify pathways in which they were likely to be active.”

The rationale for the protein-protein interaction analysis and co-expression analysis is:

“Second, we used protein-protein interaction databases (PPIDs) such as BioGRID, IntAct, Pathway Commons, and NURSA to ascertain whether co-dependent genes might interact physically with one another. Third, we searched CCLE proteomics data for proteins whose expression levels across ~375 cell lines strongly correlated with the level of each DUB; it has previously been observed that proteins in the same complex are often co-expressed to a significant degree across a cell line panel (Nusinow et al., 2020).”

– Please provide information on how thresholds were selected. For example, genes whose knockout resulted in 20 or more DE genes – why 20? Page 24: why top 7 co-dependent genes? (Sometimes, details are available in the methods section, but it would help readers to have it briefly mentioned in the main text as well.) What do the authors consider a significant "normalized enrichment score", what a good "CMap tau score"?

These are all good points. In response we have added the text to clarify threshold our selections, while also admitting that the specific values are sometimes somewhat arbitrary. Our goal was to set them at a level that emphasized strong phenotypes that we might attempt to explain biologically. For example, for co-dependent genes we played with a range of threshold values and found using a large number had little impact on the conclusions: when more weaker phenotypes were included in the analysis, few additional insights were obtained (see below). We have provided the complete dataset so other researchers interested in exploring the data more deeply can easily apply different thresholds.

There were also some boundaries imposed by the methodology. For example, the CMap tool requires at least 10 differentially expressed genes for a signature query, but not all gene names are included in the CMap analysis because the tool uses only ~10,000 genes that are either measured or judged to be reliably inferred from L1000 signatures. Thus, we selected for analysis only knockouts that resulted in >20 significantly differentially expressed genes. All raw RNA-seq data are available on GEO (accession number: GSE187008) and differential expression results are provided on Synapse (syn25008205), so the readers can easily explore subtler phenotypes as well.

To explain our choice of 7 top co-dependent genes for the DepMap analysis, we added the following sentence to the Results section: “When the top five, seven, or ten co-dependent genes were analyzed with GO enrichment analysis, the overall results were similar, but because the top seven co-dependent gene analysis yielded the largest number of anticipated GO terms for the well-characterized DUBs, we used the top seven co-dependent genes.” These details are reiterated in the Methods section. We uploaded the code for analyzing the top co-dependent genes in the DepMap to GitHub (https://github.com/lauradohertyws/DUBcodependencies) so that the reader can easily run the code with different thresholds.

We did not use the Normalized Enrichment Score as a threshold for significance. Instead, we used the adjusted p-value to identify significant gene sets (adjusted p-value < 0.05). We therefore added the adjusted p-values in parentheses along with the Normalized Enrichment Scores to demonstrate the significance of the results.

The Broad recommends that CMap tau scores larger than 90 be considered significant. The following sentence is included in the Results section:

“We used the recommended threshold of tau similarity score > 90 to determine significantly similar perturbations.”

4. Since your paper is considered for the "Tools and Resources" section, we won't make this a prerequisite for publication, but the confidence of the community in the mined data would be greatly enhanced by providing at least one targeted follow-up experiment on one of the novel hypotheses generated – for example substantiating novel findings presented in Figure 6.

We certainly appreciate this feedback but are not sure that we understand the criticism. From our perspective, the majority of the manuscript is made up of follow-up experiments designed to test the veracity of conclusions from database analyses. Admittedly, these do not have the depth one would expect for a manuscript focused on a single gene, but we have arrived at a number of new and potentially significant conclusions. The Discussion and DUB Portal now discuss how our “validation” studies could be followed up further. Specific examples of hypotheses from data-mining that were confirmed experimentally include the following:

1. By comparing the KO of USP14 to inhibition of USP14 with a small molecule inhibitor we confirmed that loss of USP14 enzymatic activity results in selective upregulation of the ubiquitin gene. This provides experimental evidence that the correlation between USP14 and ubiquitin in the DepMap is meaningful. Given the many reported functions for USP14, the resulting hypothesis that it plays a key role in ubiquitin recycling, and this is likely to be its primary non-redundant function is unexpected.

2. Using additional CRISPR knockouts we compared the impact of loss of USP8 on cytokine transcription to the impact of loss of ESCRT components to determine whether the phenotype was likely due to perturbed ESCRT function. The ESCRT members we selected to test were chosen based on the analysis of the DepMap.

3. Based on the DepMap analysis, we predicted that C16orf72 would be a novel regulator of TP53 signaling. When we knocked out C16orf72 we observed an impact on TP53 target genes. To our knowledge (at least at the time of our experiment) no connection between C16orf72 and TP53 existed in the literature.

4. Given the strong correlation between MDM2 and USP7 (as well as more recent literature on USP7), we predicted that knockdown of TP53 would rescue the impact of a USP7 inhibitor, and we tested this experimentally.

5. Analysis of the DepMap suggested that USPL1’s key function is in the Little Elongation Complex. Although USPL1 has been connected to the Little Elongation Complex in the literature, USPL1 has not been well-characterized, so the systematic comparison of USPL1 loss to loss of other components of the Little Elongation Complex is a valuable test of data derived from integrated DUB analysis. We conclude the pleiotropic phenotype associated with following loss of USPL1 is mediated by the Little Elongation Complex rather than another, as-yet undiscovered function of USPL1.

6. We also provide confidence in the mined data via systematic comparisons with a large body of existing DUB literature. This should provide confidence to readers less familiar with DUB biology.